# Integrative transcription start site analysis and physiological phenotyping reveal torpor-specific expression program in mouse skeletal muscle

Ruslan Deviatiiarov[1,2], Kiyomi Ishikawa[3], Guzel Gazizova[1], Takaya Abe[4], Hiroshi Kiyonari [4], Masayo Takahashi[3], Oleg Gusev [1,2,5,6 ✉] & Genshiro A. Sunagawa [3✉]

Mice enter an active hypometabolic state, called daily torpor when they experience a lowered caloric intake under cold ambient temperature. During torpor, the oxygen consumption rate in some animals drops to less than 30% of the normal rate without harming the body. This safe but severe reduction in metabolism is attractive for various clinical applications; however, the mechanism and molecules involved are unclear. Therefore, here we systematically analyzed the gene expression landscape on the level of the RNA transcription start sites in mouse skeletal muscles under various metabolic states to identify torpor-specific transcribed regulatory patterns. We analyzed the soleus muscles from 38 mice in torpid and non-torpid conditions and identified 287 torpor-specific promoters out of 12,862 detected promoters. Furthermore, we found that the transcription factor ATF3 is highly expressed during torpor deprivation and its binding motif is enriched in torpor-specific promoters. *Atf3* was also highly expressed in the heart and brown adipose tissue during torpor and systemically knocking out *Atf3* affected the torpor phenotype. Our results demonstrate that mouse torpor combined with powerful genetic tools is useful for studying active hypometabolism.

[1] Regulatory Genomics Research Center, Institute of Fundamental Medicine and Biology, Kazan Federal University, Volkova str.18, Kazan, Tatarstan 420012, Russian Federation. [2] Endocrinology Research Center, Dmitriya Ul'yanova str. 11, 115478 Moscow, Russian Federation. [3] Laboratory for Retinal Regeneration, RIKEN Center for Biosystems Dynamics Research, 2-2-3 Minatojimaminami-machi, Chuo-ku, Kobe, Hyogo 650-0047, Japan. [4] Laboratory for Animal Resources and Genetic Engineering, RIKEN Center for Biosystems Dynamics Research, 2-2-3 Minatojimaminami-machi, Chuo-ku, Kobe, Hyogo 650-0047, Japan. [5] Department of Regulatory Transcriptomics for Medical Genetic Diagnostics, Graduate School of Medicine, Juntendo University, Tokyo 113-8421, Japan. [6] RIKEN Center for Integrative Medical Sciences, RIKEN, 351-0198 Yokohama, Japan. ✉email: o.gusev.fo@juntendo.ac.jp; genshiro.sunagawa@riken.jp

Mammals in hibernation or daily torpor can reduce their metabolic rate to 1–30% of that of euthermic states and enter a hypothermic condition without any obvious signs of tissue injury[1,2]. How mammals adapt to such a low metabolic rate and low body temperature without damage remains one of the central questions in biology. Mammals maintain their body temperature ($T_B$) within a certain range by producing heat. In the cold, the oxygen requirements for heat production increases at a rate negatively proportional to the body size[3]. Instead of paying the high heat production cost, some mammals can lower their metabolism by sacrificing body temperature homeostasis. This condition, in which the animal reduces its metabolic rate followed by whole-body hypothermia, is called active hypometabolism[4]. As a result, the homeostatic regulation of body temperature is modified, and the total energy usage drops. This hypometabolic condition is called hibernation when it lasts for months and daily torpor when it occurs daily.

Recently, Sunagawa proposed four conditions to be required in mammalian active hypometabolism:[4] 1) tolerance to low body temperature, 2) tolerance to low oxygen consumption, 3) suppression of body temperature homeostasis, and 4) heat production ability under a low metabolic rate. Of these conditions, 1) and 2) were found to be cell/tissue-specific or local functions, which prompted researchers to analyze genome-wide molecular changes in various tissues of hibernators, including the brain, liver, heart, skeletal muscles, and adipose tissues. With the development of high-throughput sequencing approaches, such as RNA-seq and microarrays, a series of transcriptomic investigations were conducted in well-studied hibernating animals, including ground squirrels[5–8], bears[9–11], and bats[12,13]. Recent proteomics studies in ground squirrels using two-dimensional gel electrophoresis[14,15] and shotgun proteomics[16] also explored the post-transcriptional regulation of hibernation. Furthermore, several studies demonstrated epigenetic changes during hibernation[17,18], and a relationship of miRNAs in the process[19].

Due to the lack of detailed genome information in hibernators, i.e., squirrels, bats, and bears, the interpretation of high-throughput sequencing results is challenging in these animals. Instead, the mouse, *Mus musculus*, has rich genetic resources and could overcome this weakness. Notably, the mouse is well-known to enter torpor by fasting[20,21], and we have developed an improved method to initiate and detect fasting-induced torpor (FIT) in mice reproducibly[4]. Furthermore, one group identified neurons regulating the induction of FIT[22], and our group identified genetically labeled neurons that can induce a hibernation-like state in mice[23]. Such recent discoveries make the mouse a suitable and convenient animal model for studying active hypometabolism.

This study aimed to analyze the comprehensive gene expression at the skeletal muscle by introducing mice as a model for active hypometabolism, taking advantage of the rich and powerful genetic technologies available for this animal. To reconstruct genome-wide changes in gene expression, we performed Cap Analysis of Gene Expression (CAGE) in soleus muscles taken from 38 animals under various metabolic conditions. We found that entering torpor and restoring activity were associated with distinct changes in the transcriptomic profile, including marked changes in the distribution of transcription start sites within a promoter (promoter shift). We also present evidence that the torpor-specific promoters are related to the torpor phenotype by deleting the gene.

## Results

### Fasting-induced torpor shows a reversible transcriptome signature.
C57BL/6 J mice (B6J mice) enter FIT when deprived of

food for 24 h[4]. In this study, torpor was defined as having a lower $T_B$ and oxygen consumption rate ($VO_2$) than the 99.9 % credible interval of the baseline, which was defined individually for each animal. The animals return to the normal condition without any damage even after experiencing hours of extreme hypothermic and hypometabolic conditions. To analyze the reversibility in peripheral tissue gene expression during the FIT, we isolated soleus muscles from B6J mice on day 1 (Pre, $n = 4$), 2 (Mid, $n = 8$), and 3 (Post, $n = 4$) at ZT-22 as experiment #1 (Fig. 1a). We chose these time points because B6J mice usually start to enter torpor at around ZT-14, and at ZT-22, which is two hours before the light is turned on, the animals are very likely to be torpid[4]. Indeed, the $VO_2$ was higher in the Pre and Post groups and was lowest in the Mid group (Fig. 1b). Skeletal muscle is a popular tissue for hibernation research because it shows little atrophy even during prolonged immobility. Therefore, many transcriptomic and proteomic studies have been performed with skeletal muscles in the past[6,11,24–26]. Using RNA from the muscle samples, we analyzed genome-wide transcription profiles using the CAGE approach[27,28]. Based on the transcription start sites (TSS) distribution, we identified 12,862 total peak clusters, reflecting genome-wide expression profile on promoters' level with single-nucleotide precision. Among all the TSSs, 11,133 were associated with 10,617 genes, and the remaining was out of ±500-bp regions from the 5′ ends of annotated genes.

The multidimensional scaling (MDS) plot of the promoter-level RNA expression showed that the Pre and Post groups had distinct expression profiles from the Mid group (Fig. 1c, d). During torpor, the animal may show both high and low metabolism due to the oscillatory nature of this condition. Indeed, the animal in the Mid group showed a broad diversity of metabolic rates (Fig. 1b), which indicates animals are either in torpor or between torpid states. Each number in Fig. 1b, c represents the same animal in the Mid group. Despite the metabolism during torpor forming two clusters (Fig. 1b), the CAGE cluster profile did not show clustering within the Mid group according to metabolic state (Fig. 1c, d), indicating that the oscillating metabolic change during torpor does not show a clear difference in transcription.

To test the reproducibility of this experiment, we performed another independent set of samplings and CAGE analysis (experiment #2). We obtained 2, 5, and 3 samples for the Pre, Mid, and Post states, respectively. In experiment #2, the $VO_2$ at sampling showed a similar pattern as in experiment #1 (Supplementary Fig. 1a), and the MDS plot showed that the Pre and Post groups had a distinct transcriptome profile from the Mid group (Supplementary Fig. 1b, c). These results were consistent with those of experiment #1.

To gain insight into the biological process underlying the reversible expression during torpor, we analyzed differentially expressed (DE) genes on the level of promoters in the Pre to Mid and in the Mid to Postconditions. The promoters were considered differentially expressed when the false discovery rate (FDR) was smaller than 0.05. Reversibly up-regulated DE promoters were defined if they show a significant increase from the Pre to Mid (FDR < 0.05) and decrease from the Mid to Post (FDR < 0.05). Reversibly down-regulated DE promoters were similarly defined but in the opposite direction (Fig. 1e). We found 589 up-regulated and 277 down-regulated promoters (representing 481 and 221 genes) from the 12,862 total promoters, with enrichment in several distinct KEGG pathways. The top 10 enriched GO terms and KEGG pathways related to both the reversibly up- and down-regulated DE genes are shown in Fig. 1f, g. Furthermore, we found enrichment of certain motifs in the promoters with reversible expression dynamics (Supplementary Fig. 1d, e). Finally, every DE promoter was ranked in the order of the total

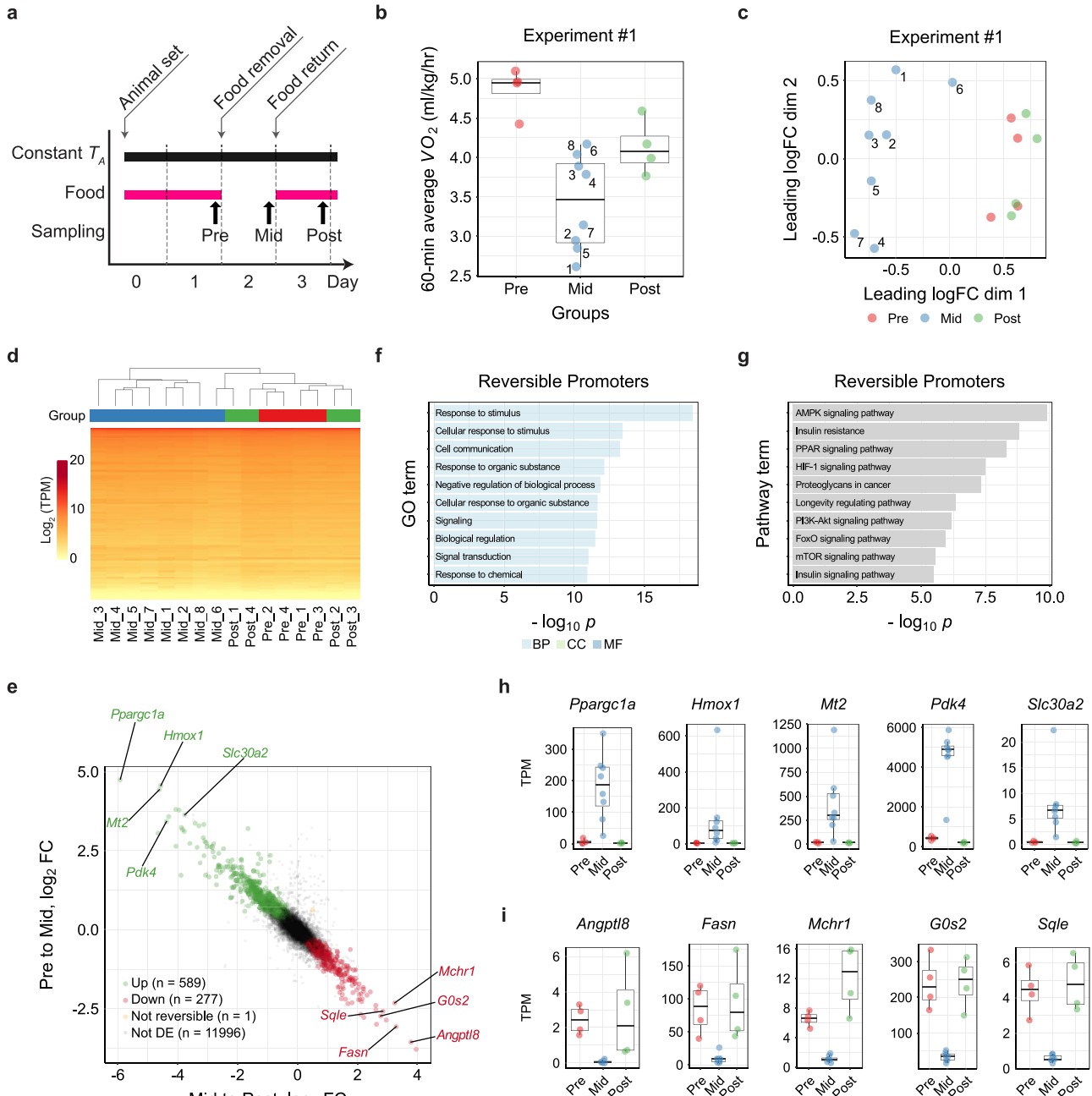

**Fig. 1 Fasting-induced Torpor Shows a Reversible Transcriptome Signature. a** Protocol for sampling muscles from Pre, Mid, and Post torpor animals to test the reversibility of the transcriptional profile of muscles during torpor. **b** Boxplots for the $VO_2$ of animals at sampling in the reversibility experiment #1. Each dot represents one sample from one animal. During torpor (Mid group), the median $VO_2$ was lower than during Pre or Post torpor. The band inside the box, the bottom of the box, and the top of the box represent the median, the first quartile ($Q_1$), and the third quartile ($Q_3$), respectively. The interquartile range (IQR) is defined as the difference between $Q_1$ and $Q_3$. The end of the lower whisker is the lowest value still within 1.5 IQR of $Q_1$, and the end of the upper whisker is the highest value still within 1.5 IQR of $Q_3$. Every other boxplot in this manuscript follows the same annotation rules. The numbers in the Mid torpor group are identification numbers of the animals. **c** MDS plot of the TSS-based distance in reversibility experiment #1. Each dot represents one sample from one animal. The Mid group clustered differently from the Pre and Post groups in the 1st dimension. The two internal groups seen in the Mid group in **d** were not evident in this plot, indicating the transient metabolic change during torpor was not correlated with transcription. **d** Hierarchical clustering heatmap based on the TPM of TSS detected in the reversibility experiment #1. The group colors are denoted in (**c**). **e** Distribution of CAGE clusters according to the fold-change (FC) in the TPM of Pre to Mid and Mid to Post torpor. The top five up- and down-regulated reversible promoters that had annotated downstream genes are shown. **f** Top ten enriched GO terms in the reversible promoters. **g** Top ten enriched KEGG pathways in the reversible promoters. **h**, **i** Top five up- and down-regulated reversible promoters ordered according to the magnitude of the TPM change. Promoters that had annotated downstream genes are shown.

fold-change, which was the sum of the fold-changes in both the Pre to Mid and the Mid to Post (Fig. 1h, i). To note, the reproducibility of this analysis was tested by comparing experiments #1 to #2, and in Pre to Mid and Mid to Post comparisons, the DE genes overlapped 80.17% and 76.82%, respectively.

Within the reversible promoters, to exclude the ones resulting from the direct effect of starvation and not the low metabolism, we further analyzed the transcriptomic profile of mouse muscles under several conditions designed to prevent the animal from entering torpor.

**Torpor prevention at high ambient temperature revealed hypometabolism-associated mRNA isoforms formed by alternative promoter usage.** Torpor can be induced by removing food for 24 h only when the animal is placed in a relatively low ambient temperature ($T_A$). We have shown that B6J mice enter torpor at a rate of 100% from the $T_A$ of 12–24 °C[4] and that confirmed that some animals do not enter torpor at $T_A$ of 28 °C (Supplementary Fig. 2a). We further tested whether the animals could enter torpor at $T_A$ of 32 °C (Supplementary Fig. 2b). In this warm condition, even if the animals were starved for 24 h, they did not enter torpor, possibly due to the lack of heat loss compared to animals at lower $T_A$s. Taking these two requirements into account, fasting and low $T_A$, we designed two torpor-preventive conditions and compared the expression in the muscles under these conditions to that under the ideal torpor state (Fig. 2a). One is a high $T_A$ (HiT) environment, and the other is a non-fasted (Fed) condition. Both conditions prevented the animals from inducing torpor because one of the two essential requirements was lacking. We then compared the tissue from these conditions to the ideal torpid tissue from fasting animals at a low $T_A$ and obtained the transcripts that were differentially expressed from torpor in each non-torpor condition. The expression differences shared in these two experiments would be those affected by both low $T_A$ and fasting, and therefore would be the essential expressions for active hypometabolism, hereafter defined as hypometabolic promoters. One aspect to note is that either 24-h fasting or low $T_A$ is an essential factor in this experiment setup, but in other environments, such as in lower $T_A$ or longer fasting, removing the factors may not prevent torpor. Therefore, the hypometabolic factors will include the most genes related to torpor, but it still includes a fraction of genes that are related only to hunger or low $T_A$.

We first compared the $VO_2$ in the HiT and Fed groups against the Mid group (Fig. 2b). Even though both groups had no animals entering torpor, the HiT group showed a lower $VO_2$ while the Fed group showed a higher metabolism. Due to the food availability, the Fed group showed high RQ compared to the Mid and HiT groups (Supplementary Fig. 2c). Next, we compared the expression profile acquired from the CAGE analysis of tissues from both groups. The MDS plot and hierarchical clustering showed that the Mid, Fed, and HiT groups consisted of independent clusters (Fig. 2c, d). This finding indicated that the expressions during torpor (Mid group) were distinct from those during starvation alone (HiT) or at low $T_A$ alone (Fed).

To extract the hypometabolic state–associated promoters, we performed the DE analysis (Fig. 2e) between the HiT to Mid and the Fed to Mid. CAGE clusters up-regulated in both the HiT to Mid and the Fed to Mid were those that were up-regulated during torpor regardless of the initial condition, i.e., warm $T_A$ or no fasting (green dots in Fig. 2e). There were 330 of these up-regulated hypometabolic promoters from a total of 12,862. On the other hand, 137 CAGE clusters that were down-regulated in both the HiT to Mid and the Fed to Mid, were promoters that were

down-regulated regardless of the initial condition, and thus were the down-regulated hypometabolic promoters (red dots in Fig. 2e). The enrichment analyses of GO terms and KEGG pathways were performed (Fig. 2f, g), and the motifs enriched in the hypometabolic promoters were also analyzed (Supplementary Fig. 2d, e). The top five promoters that had annotated genes nearby are listed as up- and down-regulated hypometabolic promoters in Fig. 2h, i, respectively.

These results showed that considerable numbers of genes are specifically involved in the active hypometabolic process and independent from either hunger or cold responses. One of these genes, *Ppargc1a*, which was found at the top of the up-regulated hypometabolic promoters, was also found at the top of up-regulated reversible promoters (Fig. 1h). Such a gene is a good candidate for a torpor-specific gene because it belongs to both the reversible and the hypometabolic groups in this study. Therefore, we next merged the results of the reversible and the hypometabolic promoters to specify the torpor-specific promoters and elucidate the fundamental transcriptional network of active hypometabolism in peripheral tissues.

**Identification of torpor-specific promoters and their dynamics.** Our analyses on two essential torpor characteristics, i.e., reversibility and hypometabolism, revealed that the skeletal muscle of torpid mice has a specific transcriptomic pattern (Figs. 1 and 2). Combining these results, we obtained torpor-specific expressed promoters, defined as the intersection of the reversible and the hypometabolic promoters. We found 226 up-regulated and 61 down-regulated torpor-specific promoters (Fig. 3a). The top five promoters ordered according to the sum of the fold-change observed in the two groups (reversible and hypometabolic promoters) are shown in Fig. 3b, c. Remarkably, "protein binding" in the molecular function category in the GO terms was listed in the top ten enriched GO terms (Supplementary Fig. 3a). This group includes various protein-binding gene products, including transcription factors. To highlight the predominant transcriptional pathway related to torpor, we ran an enrichment study of KEGG pathways with the torpor-specific promoters. We obtained 13 pathways that showed statistically significant enrichment (Supplementary Fig. 3b). In particular, the mTOR pathway, which includes various metabolic processes related to hibernation and starvation, was identified. Furthermore, we analyzed the enriched motifs in the torpor-specific promoters (Supplementary Fig. 3c, d) and found 131 significantly enriched motifs out of 579 motifs registered in JASPAR 2018[29].

CAGE analysis can detect TSSs at a single base-pair resolution, and therefore, it can be used to estimate the architecture of the promoter[30]. The shape index (SI) is one of the major indices used to evaluate promoter architecture[31]. "Narrow" promoters initiate transcription at specific positions, while "broad" promoters initiate transcription at more dispersed regions. It is widely accepted that the promoter shape differs among different tissues or conditions[32,33]. To detect promoter dynamics in the skeletal muscle under different metabolic conditions, we analyzed the promoter shape of each of the detected promoters in the reversible, hypometabolic, and torpor-specific groups (Fig. 3d). In the torpor-specific groups, the down-regulated promoters showed a significantly different shape when compared to all muscle promoters (Fig. 3e), while the GC richness did not show a difference (Fig. 3f).

The torpor-specific active promoters were identified through the expression changes in the natural cycle of FIT and two environmental cues which prevent the animals from entering FIT, namely feeding and high ambient temperature. To further narrow down causal promoters for FIT regulation, an adaptive method to prevent the animal entrance to FIT was introduced.

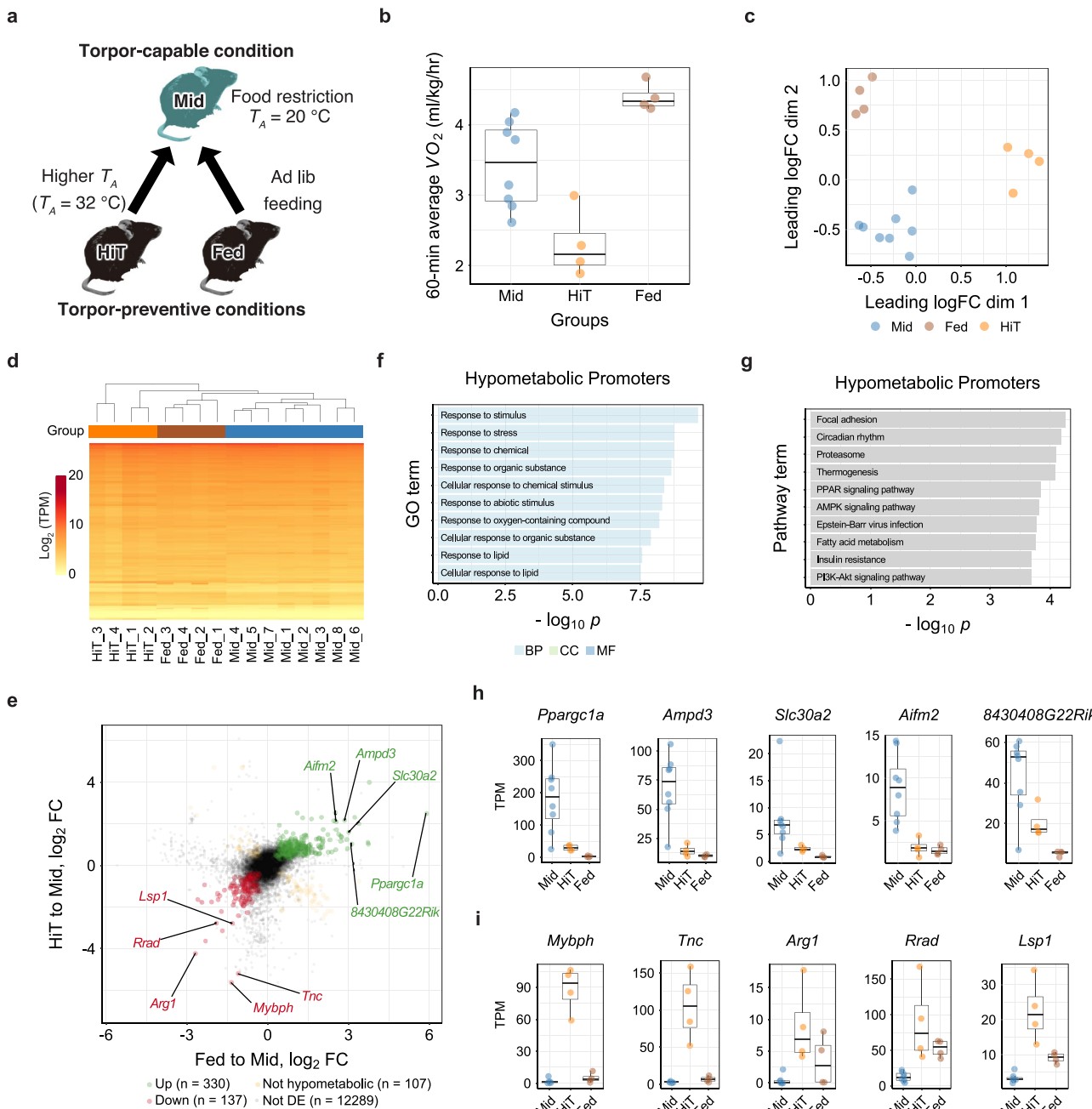

**Fig. 2 Torpor Prevention at high $T_A$ Revealed Hypometabolism-associated Promoters. a** Protocol for detecting the hypometabolic expression by sampling muscles from two groups in which torpor was prevented (HiT and Fed groups, $n = 4$ for each). For the torpid group, the samples collected in the reversibility test were used (Mid group, $n = 8$). **b** Boxplots for the $VO_2$ of animals at sampling in the hypometabolic experiment. Each dot represents one sample from one animal. During torpor prevention by high-$T_A$ (HiT group), the $VO_2$ was lower than in the Mid group, and when torpor was prevented by food administration (Fed group), $VO_2$ was higher than in the Mid group. **c** MDS plot of the TSS-based distance in the hypometabolic experiment. Each dot represents one sample from one animal. The Mid, Fed, and HiT groups were clustered separately. **d** Hierarchical clustering heatmap based on TPM of the TSS detected in the hypometabolic experiment. The group colors are denoted in (**c**). **e** Distribution of CAGE clusters according to the fold-change in TPM of the HiT to Mid and Fed to Mid groups. The top five up- and down-regulated hypometabolic promoters that had annotated downstream genes are shown. **f** The top ten enriched GO terms in the hypometabolic promoters. **g** The top ten enriched KEGG pathways in the hypometabolic promoters. **h, i** The top five up- and down-regulated hypometabolic promoters are ordered according to the magnitude of the TPM change. Promoters that had annotated downstream genes are shown.

***Atf3* is related to the regulation of FIT.** The torpor-specific promoters we found may represent regulators both upstream and downstream of the torpor transcriptional network. To further elucidate the early events involved in torpor-specific metabolism in peripheral tissues, it was necessary to place the animal in a condition where it had an extreme tendency to enter torpor and compare the muscle gene expression with regular torpor entry. For this, we mimicked the classical technique, sleep deprivation, which is frequently used in basic sleep research[34,35], and performed torpor deprivation by gently touching the animal. All of the torpor-deprived animals showed a similar metabolism level to the metabolically non-torpid animals in the Mid-torpor

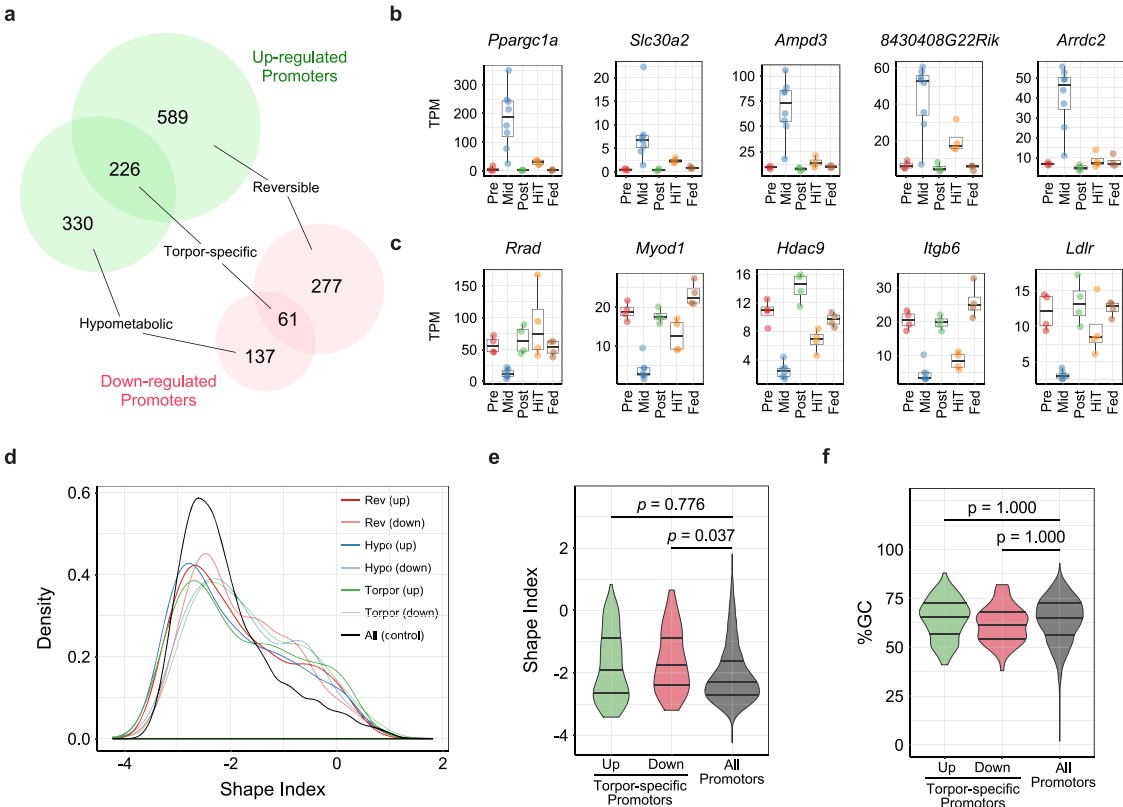

**Fig. 3 Identification of Torpor-specific Promoters and their Dynamics. a** Torpor-specific promoters were defined by the intersection of reversible and hypometabolic promoters. Up-regulated torpor-specific promoters ($n = 226$), which were CAGE clusters that were highly expressed exclusively during torpor, were at the intersection of the up-regulated reversible ($n = 589$) and hypometabolic promoters ($n = 330$). Down-regulated torpor-specific promoters ($n = 61$), which were CAGE clusters that were highly suppressed exclusively during torpor, were at the intersection of down-regulated reversible ($n = 277$) and hypometabolic promoters ($n = 137$). **b, c** Top five up-regulated (**b**) and down-regulated (**c**) torpor-specific promoters ordered according to the sum of the TPM change observed in the reversibility and hypometabolism experiments. Only promoters that had annotated downstream genes are shown. **d** Distribution of the SI of all of the mouse muscle promoters. An SI of 2 indicates a singleton-shaped CAGE TSS signal, and promoters with SI $<-1$ have a broad shape. e, f Distribution of the SI (**e**) or %GC (**f**) for torpor-specific promoters compared to all muscle promoters. The three horizontal lines inside the violin denote the 1st, 2nd, and 3rd quartile of the distribution from the upmost line.

group (Fig. 4a). Furthermore, the transcriptome profile in the muscles from torpor-deprived animals did not show a clear difference from Mid-torpor animals in MDS plots (Fig. 4b). When compared to Mid-torpor muscles, the torpor-deprived muscles had 45 up- and 27 down-regulated promoters (Fig. 4c). Among these 72 torpor-deprivation-specific promoters, one promoter starting at the minus strand of chromosome 1: 191217941, namely the promoter of the activating transcription factor 3 (*Atf3*) gene, was also found in the torpor-specific promoters (Fig. 4d). *Atf3* has two documented promoters[36]. This study detected the canonical promoter as a torpor-specific promoter and an up-regulated promoter in a torpor deprived animal. Transcripts from the other promoter located 34.7 kbp upstream from the canonical promoter did not change significantly in this study. Surprisingly, the binding site of ATF3 was one of the motifs enriched in the torpor-specific promoters (Supplementary Fig. 4a). The ATF3 motif was found in 33 of 289 torpor-specific promoters, and the peak of the motif probability was 79 bp upstream of the TSS (Fig. 4e).

To evaluate the functional universality of ATF3, the gene expression level was quantified at heart, liver, brown adipose tissue (BAT), brain, and soleus muscle during normal and FIT (Supplementary Fig. 4b). The heart, BAT and soleus muscle showed a significant increase in expression, while the liver and brain showed no difference (Fig. 4f). To test the systemic function

of ATF3 during fasting-induced torpor, *Atf3* gene was knocked out by CRISPR/Cas9-mediated gene editing. The guide sequences targeting the 5′ and 3′ regions of the *Atf3* gene were designed to delete a 12.2-kb genomic region spanning three exons of *Atf3* on chromosome 1 (Supplementary Fig. 4c). The FIT phenotype of eight F0 siblings confirmed as *Atf3* KO mice were tested. All mice entered torpor but showed higher metabolism during FIT than controls (Fig. 4g, h). Two strains—ATF3-021 and 025—out of eight F0 *Atf3* KO mice were selected and crossed with wild-type animals to produce heterozygous KO mice. The four KO alleles segregated from these two lines are shown in Supplementary Fig. 4d. The F1 heterozygous KO mice holding the identical KO allele were crossed multiple times to obtain wildtype, heterozygous, and homozygous *Atf3* KO mice. Every strain was viable, and the growth was normal (Supplementary Fig. 4e). To evaluate the gene deletion effect of *Atf3* to FIT, both the minimal $T_B$ and $VO_2$ during torpor of the KO mice were compared to those of wild-type animals (Supplementary Fig. 4f). A Bayesian statistical model including the phenotype differences as parameters was designed, and the parameters were estimated from the observed data. Both heterozygous and homozygous ATF-021a KO were estimated to have higher $T_B$ than wild-type mice (Supplementary Fig. 4g), although $VO_2$ did not have any difference. Interestingly, 025b has shown lower $T_B$ in heterozygous and homozygous strains with any differences in $VO_2$.

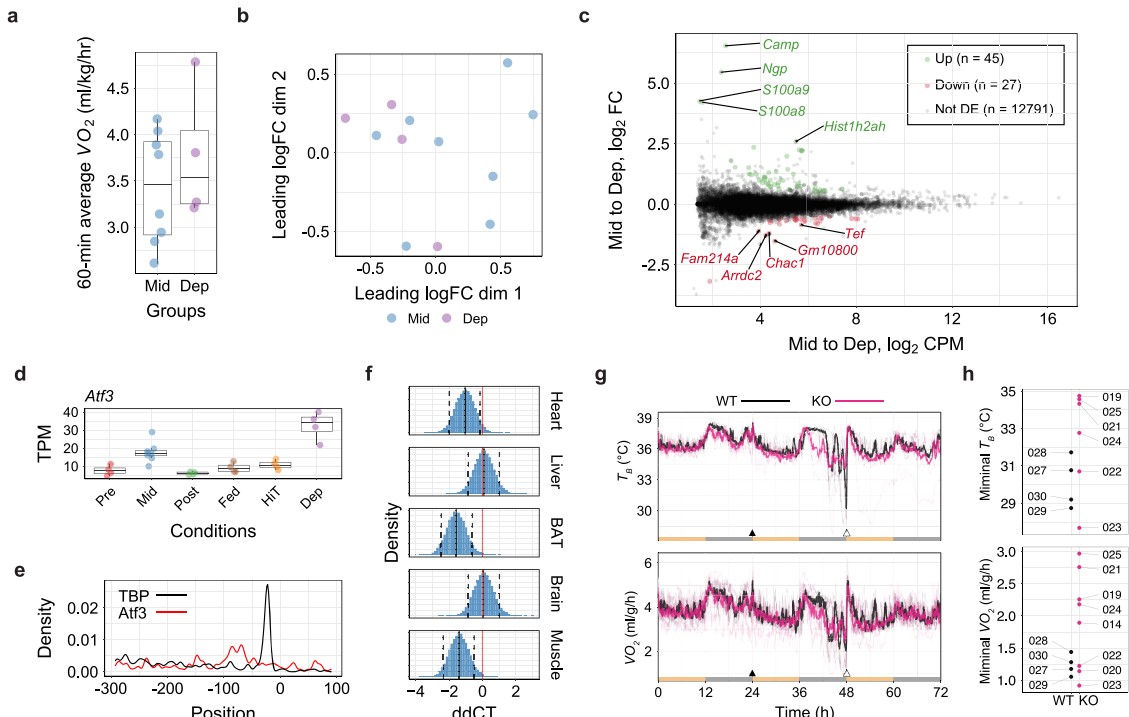

**Fig. 4 Atf3 is related to FIT regulation. a** Boxplots for the $VO_2$ of animals at sampling in the torpor deprivation experiment. Each dot represents one sample from one animal. Torpor-deprived animals (Dep group, $n = 4$) did not show an apparent change in $VO_2$ compared to the Mid group. **b** MDS plot of the TSS-based distance in the torpor-deprivation experiment. Each dot represents one sample from one animal. A clear separation between the Mid and Dep groups was not found in this analysis. **c** Distribution of CAGE clusters according to the mean TPM and the fold-change TPM of the Mid to Dep group. The top five up- and down-regulated torpor-deprivation-specific promoters that had annotated downstream genes are shown. **d** Among the torpor-specific up-regulated genes, *Atf3* was the only DE gene during torpor deprivation. **e** The motif probability of ATF3 and TBP in torpor-specific promoters. **f** Estimated ΔΔCT of *atf3* mRNA for each organ from normal to torpid condition. The black solid line and the dashed lines denote the median and the lower and upper 89% HPDI. The red line is drawn at zero. Heart, BAT, and the soleus muscle have lower CT during torpor, which indicates higher mRNA. See Supplementary Fig. 4b for raw CT counts. **g** The FIT phenotype of *Atf3*-KO mice ($n = 8$, pink lines) was compared to wildtypes ($n = 4$, black lines). The thick lines denote the average of either $T_B$ or $VO_2$ in each group, while thin lines show individual recordings. *Atf3*-KO have a tendency of higher metabolism during torpor compared to wildtypes. **h** The minimal $T_B$ and $VO_2$ during torpor were compared between *Atf3*-KO (pink) and wild-type mice (black). Two animals in the KO group failed to record body temperature due to equipment trouble. The three-digit numbers by the dots are the animal ID. Two KO lines, which showed higher metabolisms than wildtypes, ATF3-025, and ATF3-021, were selected for further evaluation.

We found that transcription factor ATF3, one of the torpor-specific genes at the soleus muscle, is highly expressed during torpor deprivation. When the gene is deleted systemically, we found two strains ATF3-021a and ATF3-025b showed shallower and deeper torpor phenotypes than wild-type animals. Collectively, the results suggest *Atf3* expression is related to the torpor phenotype, and lack of the gene changes the minimal $VO_2$ during the FIT.

## Discussion

In this study, we identified 287 torpor-specific promoters in B6J mouse skeletal muscle (Fig. 3a). Specificity was assured by including both reversible and hypometabolic promoters (Figs. 1a, 2a). The results enabled us to identify likely metabolic pathways enriched during torpor (Supplementary Fig. 3b). Although skeletal muscle is not the core tissue in mitochondrial metabolism suppression during torpor[37], the current study gained insight into transcriptional changes in muscles during daily torpor in mice.

The circadian rhythm was the most enriched KEGG pathway by torpor-specific promoters. The circadian clock is important in organizing metabolism and energy expenditure[38]. In our study, the core circadian clock gene *Per1* was up-regulated torpor-specifically, and *Arntl1* was up-regulated during torpor but not included in the hypometabolism-associated promotors. Because *Per1* and *Arntl1* are normally expressed in reversed circadian phases, our results in which both components were up-regulated

together indicated that the circadian clock was disrupted in the skeletal muscle during torpor. Several past studies have focused on the central circadian clock of hibernation[39,40] or the chronic effect of cold and hunger to the peripheral clock relative to the central clock[41]. However, little is known about the peripheral circadian clock in acutely fasted torpid animals. Thus, our results may provide evidence that the peripheral clock is disrupted at the entrance of active hypometabolism.

Similarities between fasting during hibernation or daily torpor and calorie restriction in non-hibernating mammals are reported[42]. During long-term torpor, such as in hibernating mammals, carbohydrate-based metabolism switches to lipid use. Many studies have suggested that the activation of AMPK is important in torpor induction[43–45]. However, another study demonstrated AMPK activation only in white adipose tissue, not in the liver, skeletal muscle, brown adipose tissue, or brain, during hibernation[46]. Our study corroborates the findings of Horman's research by demonstrating no significant changes in the AMPK-encoding gene expression during torpor in skeletal muscle. Although, the enriched AMPK signaling pathway without AMPK expression bespeaks a complex nature of muscle transcription in daily torpor.

The PPAR-signaling pathway also regulates lipid metabolism. Numerous studies have shown increased PPARs in various organs at the mRNA and protein levels during torpor, in several

hibernating species[42,46]. Recently, an over-expression of PPARα protein in mouse liver, comparable to that in hibernating bats, was reported, suggesting a potential hibernation capability of mice[47]. According to our data, *Ppara* is up-regulated in torpid mice muscle along with several target genes associated with cholesterol metabolism and fatty acid transport. Remarkably, the *Ppargc1a* gene, encoding PGC-1α (peroxisome proliferator-activated receptor-γ coactivator-1), was also over-expressed in mouse soleus muscle during torpor. Recently, PGC-1α activation was suggested to be responsible for protecting skeletal muscle from atrophy during long periods of torpor in hibernators[48]. Our results suggest that a similar pathway may be activated in mouse torpor as well.

We found that the insulin/Akt and mTOR signaling pathways, which have roles in skeletal muscle remodeling and metabolic rate depression, were enriched. Previous studies showed that insulin signaling is inhibited in the skeletal muscle of torpid gray mouse lemurs[49] and that the Akt kinase activity is suppressed during torpor in multiple tissues of ground squirrels[50–52]. The suppressed Akt activity is accompanied by a reduction in mTOR activation, leading to a state of protein synthesis inhibition during torpor in hibernators[52–54]. Our results demonstrated a down-regulation of *Igf1*, which encodes IGF-1, and activation of *Mtor*, which encodes mTOR, in torpor, which appears paradoxical to past studies.

The Insulin/Akt pathway also controls the phosphorylation and activation of the FOXO1 transcription factor, a disuse atrophy signature that up-regulates the muscle-specific ubiquitin ligases *Trim63* (MuRF1) and *Fbxo32* (Atrogin-1). In our study, we found that FOXO1, MuRF1, and Atrogin-1 were up-regulated, as in the case of disuse atrophy in mice and rats[55,56].

In summary, we found that the up-regulation of PGC-1α and down-regulation of IGF-1 in the skeletal muscle of torpid mice are similar to hibernating animals, in which they contribute to muscle protection and the suppression of protein synthesis. On the other hand, muscle atrophy and autophagy signatures such as FOXO1, MuRF1, and Atrogin-1 were up-regulated during torpor, indicating that atrophic changes are also progressed. Furthermore, mTOR, a signature of muscle hypertrophy, activation was found. Thus, we can conclude that mouse torpor has a unique transcription profile, sharing signatures with hibernation, starvation-induced atrophy, and muscle hypertrophy.

The CAGE technology enabled us to evaluate the dynamics of the torpor-specific gene expression on the level of promoters. We found that down-regulated torpor-specific promoters were narrower than other muscle promoters; that probably, reflects the general impact of hypometabolism on RNA transcription (Fig. 3e). To gain insight into the upstream network of torpor, we evaluated a torpor-deprived condition. Note that this dynamic state, the torpor-deprived condition, is challenging to induce in hibernators because very little stimulation can cause them to halt torpor induction. Taking advantage of this torpor-deprivation state in mice, we identified transcription factor ATF3 as a candidate factor correlated with the need to enter torpor (Fig. 4d). The altered torpor phenotype of *Atf3*-KO mice supports the torpor-related function of ATF3 (Fig. 4h and Supplementary Fig. 4e).

ATF3 is a well-known stress-inducible transcription factor, and its expression is induced by cellular stresses such as DNA damage, oxidative stress, and cell injury[57]. In addition to the stress responding aspect, numerous investigations suggest its regulatory function to cellular metabolism[58]. Both aspects of *Atf3*, namely the stress response and the metabolism regulation, may explain the role of this gene in fasting-induced torpor.

As one of the significant responses as a stress-inducible molecule, recent cumulative evidence suggests the induction of

ATF3 in ischemia/reperfusion (I/R) injuries in various organs[59–62]. In the current study, ATF3 was identified as a torpor-drive correlated factor. Torpor is an active hypometabolic condition, which shows a drastic reduction in oxygen consumption. To stay healthy under low oxygen consumption, tissues would benefit from becoming tolerant to ischemia during torpor. Therefore, we propose the hypothesis that *Atf3* is mediating the initiation of hypometabolism, and because of that, it is expressed to protect the organs under stressful conditions such as ischemia.

Another potential role of *Atf3* in the chain of torpor reaction is its metabolism pathway modifying ability. Systemic deletion of *Atf3* results in no obvious phenotypes[63]. However, series of studies suggest the ATF3 involvement in glucose metabolism in various organs and tissues. In the pancreas, *Atf3* up-regulates the expression levels of proglucagon[64] and deleting *Atf3* from the pancreas specifically results in low serum glucose[65]. In addition, pancreas- and hypothalamus-specific *Atf3* knockout shows a leaner phenotype due to decreased food intake and increased energy expenditure[61]. Collectively, *Atf3* can be assumed as a counter-reaction to systemic glucose depletion or nutrition deficiency. Therefore, the increased *Atf3* expression in the muscle, exaggerated by torpor deprivation, denotes the potential *Atf3* function for hypometabolism tolerance. One thing to note is the dissimilar phenotype of two *Atf3* knockout strains, ATF3-21a and ATF3-25b. These results indicate a clear yet complicated relationship of *Atf3* to torpor regulation, and further studies are mandatory for clarification.

In this study, torpid mouse exhibits increased *Atf3* expression in the skeletal muscle, heart, and BAT but not in the brain and liver. One possibility is that, for the brain and liver, FIT is not as stressful as to induce *Atf3* expression. This hypothesis could be tested by expression evaluation in mice facing a much severe hypometabolic condition. Another possibility is that *Atf3* may have an organ-specific function during torpor, as it has in metabolic homeostasis and cancer[58]. Understanding the organ-dependent function of *Atf3* during the FIT, a tissue-specific loss- or gain-of-function approach is required. Furthermore, the *Atf3* promoter detected in this study is one of the two documented promoters of *Atf3*[36]. Even the same proteins are produced from the transcripts, deleting the shared protein-coding region may affect the endogenous function of the *Atf3* transcript from the other promoter. To clarify this, tweaking the promoter region to evaluate the two distinct transcripts would be necessary independently. There is a report that torpor phenotype can be affected by epigenetic changes such as the nutritional experiences during the fetal period[66]. Therefore, in future studies, epigenetic influences must be taken into consideration along with the SNP analysis.

One limitation of this study is that *Atf3* was identified through the torpor precluding study by tactile stimulation. A sleep deprivation study inspired this method. To our knowledge, this is the first time to report torpor deprivation in mice. Even we have frequently observed animals trying to lower their metabolism unless we touch them during the experiment, we do not have a quantitative test whether the torpor propensity will increase by this procedure. Therefore, the higher expression of *Atf3* in torpor-deprived animals could be explained simply by the nature of this gene as a stress-inducible gene. Moreover, little effect to torpor-debt may explain the lack of robust effect to the torpor of *Atf3* knockout animals.

The overall results of this study indicate that the mouse is an excellent animal for studying the as-yet-unknown mechanisms of active hypometabolism. Understanding the core engine of the hypometabolism in torpid tissues will be the key to enabling non-hibernating animals, including humans, to hibernate. Inducing active hypometabolism in humans would be an important

breakthrough for many medical applications[1]. The benefits of using mice are not limited to technological advances in genetics but extend to the enormous potential for in vitro studies using cell or tissue culture. In stem cell biology, patient-derived stem cells represent a valuable resource for understanding diseases and developing treatments because the cells reflect the phenotype of the patient[67]. We believe, similarly, that mouse-derived stem cells or tissues will provide a unique platform for investigating strain-specific hypometabolic phenotypes in animals. Moreover, because in vitro studies can be easily extended to experiments using human cells/tissue derived from human induced pluripotent stem cells, active hypometabolism research in mouse cells/tissues is an important step toward the realization of human hypometabolism.

## Methods

**Animals**. All animal experiments were approved by the Institutional Animal Care and Use Committee of RIKEN Kobe Branch and performed according to RIKEN Regulations for the Animal Experiments. C57BL/6 J mice were purchased from Oriental Yeast Co., Ltd., Tokyo, Japan. Until the mice were used in torpor experiments, they were given food and water ad libitum and maintained at a $T_A$ of 21 °C, relative humidity of 50%, and a 12 h light/12 h dark cycle.

For the C57BL/6 J, 42 male mice were used, and the age at experiment was $8.22 \pm 0.39$ weeks old (mean ± SD) and the weight was $23.0 \pm 1.2$ g (mean ± SD).

The *Atf3* KO mice were generated with CRISPR/Cas9-mediated gene targeting by zygote electroporation. For electroporation, C57BL/6 J pronuclear stage embryos were transferred into Opti-MEM I medium containing 25 ng/μl each of crRNA1, crRNA2, crRNA3, and crRNA4, 200 ng/μl tracrRNA (FASMAC), and 250 ng/μl Cas9 protein (ThermoFisher). CUY21EDITII and LF501PT1-10 platinum plate electrodes (BEX Co. Ltd.) were used with repeated 30 V pulses (3 ms ON + 97 ms OFF) 7 times. After electroporation, the zygotes were transferred into the oviduct of pseudopregnant ICR female mice. A total of 64 F0 mice were obtained from 201 zygotes, and 8 of them were confirmed to be *Atf3* KO mice in which the *Atf3* gene locus was deleted. Four F1 heterozygous mice segregated from two *Atf3* KO founder mice, ATF3-021 and ATF3-025, were used to establish independent mutant mouse lines (021a, 021b, 025a, and 025b; Accession. No. CDB0065E-021a, 021b, 025a and 025b, respectively: http://www2.clst.riken.jp/arg/mutant%20mice%20list.html). The genotype of mice was determined by PCR with the following primers; Fwd08: TCC CGG TAT CGA GCT AAA TG, Rev01: GGG TCG AAG CAG GGA ATC AA, Rev15: CAG CAA AGG CAC GTG TCA CTA G (Supplementary Fig. 4c). PCR product sizes of wildtype and KO alleles were 747 bp (wildtype), 1255 bp (021a), 1172 bp (021b), 310 bp (025a), and 1250 bp (025b), respectively. Guide RNA sequences were designed by CRISPRdirect (https://crispr.dbcls.jp)[68]. The crRNA and tracrRNA sequences used for genome editing were synthesized as follows (FASMAC). crRNA1; GCA AGU CAC AAC AGC GAG UGg uuu uag agc uau gcu guu uug, crRNA2; AGC GAA GGA AUC GGA UCA AGg uuu uag agc uau gcu guu uug, crRNA3; GUG CCA CAC UAA CGU UUA CCg uuu uag agc uau gcu guu uug, crRNA4; tracrRNA: AAA CAG CAU AGC AAG UUA AAA UAA GGC UAG UCC GUU AUC AAC UUG AAA AAG UGG CAC CGA GUC GGU GCU.

During the experiments, each animal was housed in a temperature-controlled chamber (HC-100, Shin Factory). To record $T_B$ continuously, a telemetry temperature sensor (TA11TA-F10, DSI, New Brighton, MN) was implanted in the animal's abdominal cavity under general inhalation anesthesia at least seven days before recording. The metabolism of the animal was continuously analyzed by respiratory gas analysis (ARCO-2000 mass spectrometer, ARCO system, Kashiwa, Japan). The animal was monitored through a networked video camera (TS-WPTCAM, I-O DATA, Inc.). This video camera can detect infrared signals, which made it possible to monitor the animal's health during the dark phase without opening the chamber.

**Daily torpor induction experiment**. Each daily torpor induction experiment was designed to record the animal's metabolism for three days unless the tissues were sampled on day 2. The animals were introduced to the chamber the day before recording started (day 0). Food and water were freely accessible. The $T_A$ was set as indicated on day 0 and kept constant throughout the experiment. A telemetry temperature sensor implanted in the mouse was turned on before placing the mouse in the chamber. The standard experimental design was as follows: on day 2, ZT-0, the food was removed to induce torpor. After 24 h, on day 3, ZT-0, the food was returned to each animal. In the torpor-prevention experiment with food administration (Fig. 2a), the food was not removed on day 2. In the torpor-deprivation experiment, one experimenter monitored the $VO_2$ and touched the mouse gently when the $VO_2$ started to drop. The metabolism monitoring for torpor deprivation was started at ZT-17 on day 2 and maintained until the mouse tissue was sampled at ZT-22.

**Body temperature and oxygen consumption modeling for daily torpor detection**. To model the temporal variation of $T_B$ and $VO_2$, we constructed the models in a Bayesian framework. From the first 24-h recordings of $T_B$ and $VO_2$ for each animal, we estimated the parameters using Markov Chain Monte Carlo (MCMC) sampling by Stan[69] with the RStan library[70] in R[71]. The method was described previously[4] and modified with software updates. For each animal, $T_B$ and $VO_2$ were evaluated every six minutes for three days. From the recordings of day 1, the baseline metabolism was estimated with a certain credible interval (CI). In this study, we used the 99.9% CI of the posterior distribution of the estimated metabolism to detect outliers. When the value is lower than the CI, that time point is defined as torpor due to an abnormally low metabolic status. When both $T_B$ and $VO_2$ met the criteria in the second half of the day, the time point was labeled as torpor. When $T_B$ or $VO_2$ was unable to record from equipment troubles, either one successfully recorded was used for torpor detection.

**Parameter estimation of Atf3-KO phenotypes**. The effect of *Atf3* deletion to the FIT phenotype was evaluated by the difference of minimal $T_B$ and $VO_2$ of *Atf3*-KO mice from the wild-type mice. A Bayesian statistical model including the strain and deletion allele information was produced, and the parameters were estimated from the observed data. Model fitting was performed using Hamiltonian Monte Carlo with its adaptive variant, the no-U-turn sampler, as implemented in version 2.12.2 of Stan with the RStan library in version 4.0.5 of R[71]. We assessed convergence by inspecting the trace plots, Gelman-Rubin R̂, and an estimate of the effective number of samples. The model priors were defined as weakly informative and conservative, which are specified in the following sections. The fundamental principles and techniques for designing the statistical models were based on the book Statistical Rethinking[72].

The minimal $T_B$ and $VO_2$ of the animals were modeled on the assumption that each genotype has a unique phenotype. Four types of *Atf3*-KOs were tested in this experiment. Each knockout has three combinations of alleles, namely the wildtype, heterozygous, and homozygous KOs. When N is the total number of the animals, and $Y_{NORMAL}$ is the minimal $T_B$ or $VO_2$ during the normal state, $Y_{NORMAL}$ of mouse $i$ can be described as the sum of the global mean parameter $\alpha$ and the group parameter $\beta$, with the noise modelled in a Normal distribution of a scale parameter $\sigma_{NORMAL}$ as:

$$Y_{NORMAL[i]} \sim Normal(\alpha + \beta_{gene[i],allele[i]}, \sigma_{NORMAL}) \\ i = 1 \cdots N \tag{1}$$

When another group parameter $\gamma$ is given as the difference of $T_B$ or $VO_2$ during torpor from the normal state, $Y_{TORPOR}$ can be described as:

$$Y_{TORPOR[i]} \sim Normal(\alpha + \beta_{gene[i],allele[i]} + \gamma_{gene[i],allele[i]}, \sigma_{TORPOR}) \\ i = 1 \cdots N \tag{2}$$

Both group parameters were sampled as

$$\beta \sim Normal\left(0, \sigma_\beta\right) \tag{3}$$

$$\gamma \sim Normal\left(0, \sigma_\gamma\right) \tag{4}$$

where $\sigma_\beta$ and $\sigma_\gamma$ were sampled from a Half-Cauchy distribution with a location parameter 0, and scale parameter 2.5. All other $\sigma$s were sampled from standard half-normal distributions with scale parameter 10. The phenotype difference of KO strains was evaluated by the posterior distribution of the difference in $\gamma$ between homozygous and wildtype, or heterozygous and wildtype (Supplementary Fig. 4g).

**Tissue sampling and RNA isolation**. Dissected soleus muscles were rapidly frozen in liquid nitrogen. The RNA was isolated using an RNeasy Fibrous tissue kit (Qiagen) according to the manufacturer's instructions. The quality of the total RNA was evaluated using a Bioanalyzer 2100 (Agilent). The quantity and purity of the RNA were estimated using a NanoDrop Spectrophotometer. The lateral or both soleus muscles were used according to the total amount of RNA needed.

**Genotyping of knockout mice by sequencing**. The genomic DNA of *Atf3* KO mice was prepared from their tail using Maxwell 16 Tissue DNA Purification Kit (Promega). Each target gene sequence was amplified by PCR with GoTaq (Promega) by the primers Fwd08 and Rev15, and sequenced. For ATF3-21a, two additional primers were used to confirm the sequence; Fwd11: CTG GAA CTG GAG TTT CAG AG and Rev18: ATG GGT CAG CAG TTT ACA A.

**Quantification of mRNA levels**. cDNA was synthesized from isolated RNA using a High-Capacity cDNA Reverse Transcription Kit (4368814, Applied Biosystems) according to the manufacturer's instructions. qPCR was performed by TaqMan Fast Universal PCR Master Mix (2X), no AmpErase UNG (4352042, Applied Biosystems) with gene-specific TaqMan probes using the StepOne Real-Time PCR Systems (Applied Biosystems). The following TaqMan probes were used: *Atf3* Mm00476033_m1, *Gapdh* Mm99999915_g1.

**Non-amplified non-tagging Illumina cap analysis of gene expression (nAnT-iCAGE) library preparation and sequencing**. Transcriptomics libraries were prepared according to a standard protocol for the CAGE method using 5 μg of extracted total RNA from mouse muscles[73]. The RNA was used as a template for the first-strand cDNA synthesis, which was then biotinylated at the 5′-end to allow streptavidin capture. Linkers were then attached at the 5′ and 3′ ends, and the second strand cDNA was synthesized. The quality of the libraries was verified using a Bioanalyzer 2100 (Agilent), and the yield was validated by qPCR. The single-end libraries were then sequenced on a NextSeq platform (Illumina) or a HiSeq 250 platform using Rapid Run mode (Illumina) in experiments #1 and #2.

**Mapping, peaks calling, and annotation**. Sequenced reads were trimmed and mapped on the mouse mm10 genome assembly using bwa and hisat2[74,75]. For each sample, we obtained CAGE-defined TSSs (CTSSs) according to the reads abundance and then clustered them using PromoterPipeline[76], the highest peaks were annotated as TSSs. These CAGE clusters were then associated with their closest genes using the Ensembl and Refseq transcripts annotation available for mm10.

**Data processing**. Data were processed in R[71] unless otherwise noted. The expression level of the 12,862 defined CAGE clusters was normalized by a sample in TPM (tags per million) and then analyzed with the edgeR package[77] with TMM (trimmed mean of M-value) normalization. For MDS (multidimensional scaling) plots, DE (differential expression), and GO and KEGG pathway enrichment analysis, several R packages were applied, including edgeR, clusterProfiler[78], and pathview[79]. Muscle enhancers were predicted de novo by applying the FANTOM5 protocol[80] to our mouse CAGE data and masked with ±500-bp regions from the 5′ ends of annotated genes. The DE results (reversible, hypometabolic, and torpor-deprivation-specific promoters) along with torpor-specific promoters are listed in Supplementary file 1.

**Basic promoter features analysis**. Promoter region features were analyzed in terms of GC content and SI[31]. The SI and %GC were calculated for ±50 bp regions around the TSS position. CpG island muscle promoters were defined by searching for overlaps with the UCSC annotation using bedtools v2.25.

**Motif analysis**. Transcription factor binding sites (TFBS) were predicted in −300/+100 bp regions around the TSS position using MEME Suite 4.11.2 and the JASPAR CORE motif library for vertebrates 2016. The position-dependent enrichment of these motifs was performed by the CentriMo tool.

**Statistics and reproducibility**. Statistical analyses were performed using R. In the CAGE analysis, genewise negative binomial generalized linear models were used for differential expression analysis in edgeR. Multiple comparison adjustment was performed using Benjamini - Hochberg correction. Motif enrichment P-values were defined by using Fisher's exact test with Bonferroni correction in CentriMo. To test the reproducibility of the CAGE analysis, we created CAGE libraries from two independent batches of mice. For animal phenotyping, Bayesian statistical models including the parameters of interest were designed, and the parameters were estimated from the observed data.

**Reporting summary**. Further information on research design is available in the Nature Research Reporting Summary linked to this article.

## Data availability

The data supporting this study are available within the paper and its supplementary files. CAGE data from this paper is available from NCBI GEO with accession number GSE117937. CAGE samples access and visualization also available in Zenbu browser http://fantom.gsc.riken.jp/zenbu/gLyphs/#config=ylDd70XVLdPufetrnXzQkB. The table of differentially expressed promoters are provided as Supplementary Data 1. Any remaining information can be obtained from the corresponding authors upon reasonable request.

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

## Acknowledgements

We thank the LARGE, RIKEN BDR for housing the mice. This work was supported by the RIKEN Special Postdoctoral Researcher program (G.A.S.), by Grant-in-Aid for Scientific Research on Innovative Areas (Thermal Biology) 18H04706 from MEXT (G.A.S.), by JSPS KAKENHI Grant-in-Aid for Scientific Research (A) (19H01066) (M.T. and G.A.S.) and by the Ministry of Science and Higher Education of the Russian Federation (agreement no. 075-15-2020-784; R.D., G.G., and O.G.).

## Author contributions

O.G., M.T. and G.A.S. designed the study. K.I. and G.A.S. performed the animal experiments and tissue sampling, supervised by G.G. R.D. and G.A.S. analyzed the data. R.D., G.G., G.O. and G.A.S. wrote the manuscript. All authors discussed the results and commented on the paper text.

## Competing interests

The authors declare no competing interests.
