## [Transparent Peer Review File · Communications Biology]

Reviewers' comments:

Reviewer #1 (Remarks to the Author):

The manuscript by Deviatiiarov et al. covers a large amount of experimental data aimed at identification of hypometabolism related gene expression changes in soleus muscle in the fasting mouse. To this end, the authors used CAGE-seq to quantify levels of capped 5' mRNA sequences, which were used to identify simultaneously gene expression and regulatory promoter motifs. They compared hypometabolism propensity of two closely related mouse strains and performed CAGE-Seq on the strain showing the largest reduction in metabolism, with hypometabolism specific promoters being identified by contrasting changes with those of mice in which hypometabolism was precluded by AdLib feeding and fasting at high ambient temperature. The authors thus define 287 hypometabolism specific promoters, of which the large majority was upregulated. GO and KEGG analysis shows a large overlap with pathways previously described [in part] or implicated in hibernators (e.g. AMPK, PI3K/Akt/mTOR, autophagy, PPAR). Following SNP comparison between strains with high and low hypometabolism propensity, and precluding mice from torpor by a tactile stimulus, a transcription factor from the CREB family, ATF3, was identified as a possible candidate. Total body knockout of ATF3 appeared to reduce the hypometabolic response to fasting. The authors conclude that the ability to deploy advanced genetic techniques makes mouse a valuable tool to explore hypometabolism mechanisms.

Although experiments are generally well-designed, a major shortcoming of the manuscript is the alignment of data of CAGE-seq analyses on B6J with the metabolic experiments on B6N. The authors tried to explain differences in hypometabolism propensity of the two strains by searching for differences in SNPs in promoters identified as hypometabolism-specific in B6J, yet this strategy did not produce significant results. This issue might be solved most easily by performing CAGE-seq on B6N, which would have the advantage of providing additional information on the importance of torpor-related pathways. Alternatively, the authors may choose to leave out or reduce the sections dealing with B6N.

A second issue is the use of soleus muscle as representative peripheral tissue related to hypometabolism propensity, motivated by the similar Tr of B6N and B6J (Fig. 1j). Firstly, Tr in B6N is similar in euthermia and torpor (Fig. 1h), whereas previous analysis on B6J shows a clear lowering of Tr during torpor in B6J (Fig. 5d of reference 4), signifying a difference in Tr regulation of both strains during torpor and refuting the statement in lines 137 ff. How was Fig. 1j constructed? Secondly, current data indicate that – particularly in daily hibernation – skeletal muscle mitochondria function does not change upon entering torpor (see work of Staples, Jastroch and others, e.g. reviewed in <https://rdcu.be/ccyzU>), although so far not established in mouse. These studies suggest liver as the leading organ in metabolic reduction in daily hibernation. Essentially, we do not know, and the current study cannot answer either, whether metabolism of mouse soleus muscle changes at all during hypometabolism. The interpretation of the results in the frame of hypometabolism is hence unsubstantiated.

As a follow-up to the above, I am not convinced that ATF3 knockout affects torpor depth or propensity (e.g. Fig. 5A). What was the fraction of knockout animals that entered torpor? Could you provide a formal analysis of differences? In that perspective, it is interesting that liver ATF3 levels do not appear to be changed during torpor.

The authors used tactile stimulation to preclude the animal to enter torpor (which failed: Fig. 5a). They suggest that torpor has a similarity to sleep in a sense that torpor propensity increases in animals where torpor is prevented. This may well be true for deep hibernators, but to my knowledge this is unproven for mouse/daily hibernators. This might be tricky as mouse torpor is strictly dependent on the circadian rhythm – even in multiday fasting, mice only spend time in torpor during the second half of dark and first half of the light phase.

On the issue of circadian rhythm, the authors state that little is known about peripheral clocks. Please see <https://doi.org/10.1073/pnas.1413135111>, providing evidence for cold and hunger inducing an advanced phase angle in peripheral tissue relative to SCN in mouse.

A striking and important finding of the current study is the death of B6N during 24h fasting under the lowest Ta and the absence of torpor under high Ta. These data may provide a significant clue what is needed to enter into or to 'survive' torpor. What do Tb and VO2 patterns in these animals look like? Were they lower in body weight? When did they die? Is death due to not being able to maintain Tb, which may relate to the absence of shifting of Tr (a Q10 lowering of VO2)? Or is death acutely at normal or slightly lower Tb? Similar questions arise for animals not entering torpor at higher temperatures.

Variation in torpor propensity does not necessarily depend on genetic factors, a common theme articulated in the manuscript, but may also involve epigenetic regulation (see e.g. doi: 10.1242/jeb.171983). Epigenetic changes, for instance DNA methylation of CpG in promoters, will also affect CAGE-Seq findings. Please point this out in the discussion.

The authors define torpor as a drop in metabolic rate below 30% - this is a value not reached in the mice examined, notably B6J (e.g. line 117, Fig 2b). Consequently, many in the field will not regard the hypometabolism phase as torpor - this should be dealt with.

Minor

24: 'is' is lacking

26: ATF3 is not highly expressed in liver, but in BAT

37: superscript of 4; 'As a result,...' suppression of thermoregulation is not a result of hypometabolism (we don't know)

53: references 17 and 18 document epigenetic changes during hibernation (which might be outside promoters or CpGs), but do not prove strict epigenetic control

54: Similarly, reference 19 examines a poikilotherm marine animal quite distant from mammalian hibernators and both reference 19 and 20 do not prove 'a role of MiRNAs', but merely associated changes.

75: Many readers may not be familiar with 'promotor shapes' (at this point) -> distribution of transcription start sites within a promoter (or the like)

90: may I suggest replace 'lower than' by: below the 99.9% CI (or the like)

147: form -> from

152: they show -> it shows

167: I think you cannot infer from the results that metabolism oscillation during torpor is an transcription-independent phenomenon; your sampling timing seems not apt and sample number too low.

169: I do not get that you exclude circadian influences by experiment #2

186: It seems that some of the most extremely changed dots in Fig. 2e are not annotated and represented in Fig. 2h,i. Can you explain?

202: the two essential -> one of the two essential

223: please provide the number of promoters

229: both hunger and cold -> either hunger or cold

259: In Fig. 4d, torpor-down is very similar to hypo-down. Yet in Fig. 4e only torpor down is shown. Please phrase an overall conclusion of Fig. 4d.

347: "reversible experiment": not sure what you mean.

359: Sure there may be no change in AMPK subunits, yet pathway analyses consistently show it among the (top) regulated pathways; how to interpret this finding? AMP/ATP ratio - are regulated genes all downstream?

Fig. S1d: the text B6N is behind the grid.

Reviewer #2 (Remarks to the Author):

The studies by Deviatiiarov and colleagues aim to establish a difference in torpor between two closely related but genetically distinct mouse strains, and then exploit those differences to identify transcriptional start sites that might be involved in the regulation of torpor. Through various comparisons, the manuscript identifies 287 "torpor-specific" promoters in soleus muscle. The authors then focus on one gene, *Atf3*, which is involved in the stress response. Knocking out *Atf3* modestly weakens the torpor phenotype in mice.

It appears that a lot of data were generated in these studies, which will be of use to the scientific community. However, it is difficult to determine which experiments were performed at the same time. For the comparisons to be valid, it is essential that the treatment groups for each experiment have been analyzed concurrently and using comparable mice (of the same body weight, age, and sex). These variables are known to affect torpor but are not accounted for in this manuscript and may confound the major findings. These concerns would need to be addressed in order to substantiate the conclusions of the paper.

Major points:

1. The analyses of the torpor phenotype in BL/6J vs. BL/6N mice do not appear to be performed at the same time and appear to have different body weights. This means that the strain difference may be confounded by experimental conditions or body weight. This is a major finding of the paper, but the experimental design does not adequately support the conclusion.
2. Lines 169-174 describe an analysis that was used to exclude possible circadian effects. The results are reported to be similar with respect to the VO₂ pattern and "a distinct transcriptome profile" but the promoters were not reported. Please provide a full analysis and discuss any differences in the results. Does the overlap among the different analyses change? Do other candidates arise?
3. The analyses comparing HiT and ad lib fed mice to food-restricted "Mid" mice do not appear to be performed at the same time, which introduces many confounds. Additionally, the manuscript does not specify how do ad lib fed mice differ from "Pre" mice. Are the mice analyzed at similar body weight, sex, age, time of day?
4. Because FIT requires cold temperature and fasting, the idea to remove one variable is potentially informative. However, the design does not test for the isolated effect of fasting (which requires fed or fasted mice BOTH concurrently housed at high ambient temperature) or the effect of temperature (which requires two concurrent groups of fed mice housed at low or high ambient temperatures). As designed, the analysis is confounded by time, does not alter the dependent variables one at a time, and does not report information on other variables that are known to modulate torpor (e.g. body weight, sex, age). Additionally, this approach assumes that the effects of these variables is additive. Is there any evidence for that? At the very least, this should be explained as a major caveat.
5. The involvement of the *Atf3* transcription factor in the regulation of torpor is unclear. The authors show that *Atf3* expression is altered during torpor and that *Atf3* may regulate torpor-specific promoters. However, it is unclear how the strain differences are explained by *Atf3*. Are there strain differences in *Atf3* regulation or expression? What is the phenotype of the *Atf3* KO mice beyond FIT?

Are there body weight or other phenotypes that might confound the FIT response? The authors should also discuss how the function of Atf3 in the stress response might relate to torpor. How does stress alter FIT or torpor? Might the KO FIT phenotype reflect an interaction with stress?

Minor points:

1. The manuscript does not adequately explain H or T(R) (lines 120-122). It is insufficient to cite the previous publication. Please explain the variables, what they indicate about the physiology of torpor, and, most importantly, what is the implication of a strain difference in H?
2. The abstract and first sentence of the introduction mention that VO₂ drops to <30% of baseline. It does not appear that any of the FIT responses in the main figures reach this threshold. The definition of torpor should be consistent in the text and figures (and the literature).
3. In line 41, "resistance" appears twice. Perhaps "tolerance" would be more appropriate because the mice do not resist (instead they actively induce) the decreases in body temperature or oxygen consumption.
4. Line 47 mentions a₂-macroglobulin and cites previous work. This molecule is never mentioned again. Was it identified in any of the current studies?
5. Line 64 claims that FIT was developed by the authors. Please see "Fasting-induced torpor in *Mus musculus* and its implications in the use of murine models for human obesity studies" published by Webb, Jagot, and Jakobson in 1982.

Reviewer Comments, Author Responses, and Manuscript Changes

The box is the comments from the reviewers.

Blue letters denote the updated text in the manuscript.

Reviewer #1

General comments

The manuscript by Deviatiiarov et al. covers a large amount of experimental data aimed at identification of hypometabolism related gene expression changes in soleus muscle in the fasting mouse. To this end, the authors used CAGE-seq to quantify levels of capped 5' mRNA sequences, which were used to identify simultaneously gene expression and regulatory promotor motifs. They compared hypometabolism propensity of two closely related mouse strains and performed CAGE-Seq on the strain showing the largest reduction in metabolism, with hypometabolism specific promoters being identified by contrasting changes with those of mice in which hypometabolism was precluded by AdLib feeding and fasting at high ambient temperature. The authors thus define 287 hypometabolism specific promoters, of which the large majority was upregulated. GO and KEGG analysis shows a large overlap with pathways previously described [in part] or implicated in hibernators (e.g. AMPK, PI3K/Akt/mTOR, autophagy, PPAR). Following SNP comparison between strains with high and low hypometabolism propensity, and precluding mice from torpor by a tactile stimulus, a transcription factor from the CREB family, ATF3, was identified as a possible candidate. Total body knockout of ATF3 appeared to reduce the hypometabolic response to fasting.

The authors conclude that the ability to deploy advanced genetic techniques makes mouse a valuable tool to explore hypometabolism mechanisms.

Reviewer #1

Although experiments are generally well-designed, a major shortcoming of the manuscript is the alignment of data of CAGE-seq analyses on B6J with the metabolic experiments on B6N. The authors tried to explain differences in hypometabolism propensity of the two strains by searching for differences in SNPs in promoters identified as hypometabolism-specific in B6J, yet this strategy did not produce significant results. This issue might be solved most easily by performing CAGE-seq on B6N, which would have the advantage of providing additional information on the importance of torpor-related pathways. Alternatively, the authors may choose to leave out or reduce the sections dealing with B6N.

Response

Thank you for raising an issue in the most fundamental part of our manuscript. Our main interest in this study was whether the expression and regulation of torpor-related genes are linked with genome differences in the two strains B6J and B6N. Unfortunately, we could not confirm such differences tied to phenotype differences in these two strains;

however, we believe that the data we have collected for this study disclosed novel information to the torpor research field. We admit that CAGE-seq analysis on B6N soleus muscle will add insight into the major question. However, from the funding perspective, the experiment will be our top priority in the future, but not for this study. For that reason, we would like to keep the B6N torpor phenotype data in the manuscript without adding the CAGE-seq of B6N.

Reviewer #1

A second issue is the use of soleus muscle as representative peripheral tissue related to hypometabolism propensity, motivated by the similar T_r of B6N and B6J (Fig. 1j). Firstly, T_r in B6N is similar in euthermia and torpor (Fig. 1h), whereas previous analysis on B6J shows a clear lowering of T_r during torpor in B6J (Fig. 5d of reference 4), signifying a difference in T_r regulation of both strains during torpor and refuting the statement in lines 137 ff. How was Fig. 1j constructed?

Response

Thank you for pointing out the ambiguity of our original manuscript. In the past paper (ref 4), T_r of B6J was estimated by $T_A = 8$ to 24. In this paper, $T_A = 28$ data was added to gain preciseness of T_r estimation (Sup Fig. 1e). As a result, the T_r of B6J was re-estimated (Sup Fig. 1f), and it turned out that B6J does not have a T_r drop (Sup Fig. 1f). Even we attempted to mention all of the points above in the original **Results** and **Methods** section (lines 125–126, 444–446), we have modified the expression to make it clearer to the reader as follows: “To compare these parameters with B6J mice, we newly recorded the $T_A = 28$ °C data missing in our previous study and recalculated both the H and T_R for B6J mice using all of the data at T_A s of 8, 12, 16, 20, 24, and 28 °C (Supplementary Fig. 1e, f)” (lines 126-128).

Fig 1j is constructed from the estimated distribution of T_r of both B6J and B6N. Although according to Reviewer #1’s response, we realized that our description was not sufficient, so we added descriptions for ΔT_r as well as ΔH as “... B6N mice had a larger H than B6J (ΔH , the difference in H during torpor for B6N versus B6J, ...” (lines 130-131) and “Interestingly, ΔT_r , the difference in T_r during torpor for B6N versus B6J, ...” (line 135).

Reviewer #1

Secondly, current data indicate that – particularly in daily hibernation – skeletal muscle mitochondria function does not change upon entering torpor (see work of Staples, Jastroch and others, e.g. reviewed in <https://rdcu.be/ccyzU>), although so far not established in mouse. These studies suggest liver as the leading organ in metabolic reduction in daily hibernation. Essentially, we do not know, and the current study cannot answer either, whether metabolism of mouse soleus muscle changes at all during hypometabolism. The interpretation of the results in the frame of hypometabolism is hence unsubstantiated.

Response

Thank you for providing us the big picture of the current understanding of skeletal muscle metabolism during torpor. From the past mitochondria studies, we are aware that metabolism reduction in muscles may not be the core of systemic hypometabolism. At the same time, our study was partially inspired by recent reports about the active involvement of transcriptome changes in skeletal muscles of hibernators, including suggestions of existing mitochondria protective mechanisms during the hibernation (Miyazaki et al., 2019 PlosONE; Wang et al., 2019 J Cell Physiol; Zhang et al., 2020 Cells, etc.). Therefore, we added the explanation about the limitation of this study due to analyzing skeletal muscle solely: “**Although skeletal muscle is not the core tissue in terms of mitochondrial metabolism suppression during torpor⁵³, the current study gained insight into transcriptional changes in muscles during daily torpor in mice**” (lines 363-365). We have a plan to evaluate the mRNA expression and the metabolite changes in the liver in future experiments.

To note, we realized that when the up-regulated hypometabolic-specific genes are fed into the GO analysis, the cellular component GO term ‘Mitochondrion’ is highly ranked. This result indicates that mitochondria may have a specific function in mouse daily torpor. We also want to point, that Fig 3f is the enrichment study for both the up- and down-regulated genes, which is why the listed genes are very different from the one below.

Reviewer #1

As a follow-up to the above, I am not convinced that ATF3 knockout affects torpor depth or propensity (e.g. Fig. 5A). What was the fraction of knockout animals that entered torpor? Could you provide a formal analysis of differences? In that perspective, it is interesting that liver ATF3 levels do not appear to be changed during torpor.

Response

Thank you for raising the fundamental question in our study. We think that ATF3 knockout does affect torpor, although the effect is not so strong. In the first *Atf3*-KO F0 screening, eight knockout animals were phenotyped, and every mouse entered torpor. However, the hypometabolism phenotype characterized by the minimal TB and VO₂ was higher than wildtype animals (Fig. 5g and 5h). We added the fraction of animals that entered torpor in the manuscript as “All mice entered torpor but showed higher metabolism during FIT than controls (Fig. 5g, h).” (lines 325–326). We also noticed that one animal (animal id 014) was erroneously missing from the lower panel of Fig. 5h, which was corrected. The figure legend was updated accordingly.

We also added the discussion about the unchanged expression of *Atf3* levels at the liver: “In this study, torpid mouse exhibits increased *Atf3* expression in the skeletal muscle, heart, and BAT but not in the brain and liver. One possibility is that, for the brain and liver, FIT is not as stressful as to induce *Atf3* expression. To test this, the expression must be evaluated in mice facing a much severe hypometabolic condition. Another possibility is that *Atf3* may have a totally organ-specific function during torpor, as it has in metabolic homeostasis and cancer⁸⁰. To understand the organ-dependent function of *Atf3* during FIT, a tissue-specific loss- or gain-of-function approach is required.” (lines 448–454).

Reviewer #1

The authors used tactile stimulation to preclude the animal to enter torpor (which failed: Fig. 5a). They suggest that torpor has a similarity to sleep in a sense that torpor propensity increases in animals where torpor is prevented. This may well be true for deep hibernators, but to my knowledge this is unproven for mouse/daily hibernators. This might be tricky as mouse torpor is strictly dependent on the circadian rhythm – even in multiday fasting, mice only spend time in torpor during the second half of dark and first half of the light phase.

On the issue of circadian rhythm, the authors state that little is known about peripheral clocks. Please see <https://doi.org/10.1073/pnas.1413135111>, providing evidence for cold and hunger inducing an advanced phase angle in peripheral tissue relative to SCN in mouse.

Response

Thank you for pointing out the previous studies about fasting induced-torpor and the circadian rhythm. We would like to respond to the results in Fig. 5a first and then to the issue of the circadian rhythm.

First, Fig. 5a shows that torpor-deprived animals do have a similar VO₂ compared to torpid animals. This might be the case if both groups have the same distribution, but the mid-torpor group has a bi-modal distribution in nature, as is explained as “During torpor, the animal may show both high and low metabolism due to the oscillatory nature of this condition” in lines 168 to 169 (Fig. 2b). Therefore, it is important to apprehend that torpor-deprived animals show higher VO₂ than the truly torpid animals in the mid-torpor group. To clarify this, we have updated the description of Fig. 5a to “All of the torpor deprived animals showed a similar level of metabolism to the metabolically non-torpid animals in the Mid-torpor group” (lines 308–309).

Second, as Reviewer #1 pointed out, chronic cold and hunger advance the phase of peripheral tissue relative to SCN (van der Vinne, Proc. Natl. Acad. Sci. U. S. A. , 2014). We believe that in our study, the animals have confronted hunger acutely (less than 24 hours), the effect from the central-peripheral clock relationship is minimized. Although we think that referring to this issue may deepen the discussion of the study, we updated the discussion of the circadian rhythm and torpor as follows: “Several past studies have focused on the involvement of the central circadian clock of hibernation^{55,56}, or the chronic effect of cold and hunger to the peripheral clock relative to the central clock⁵⁷. However, little is known about the peripheral circadian clock in acutely fasted torpid animals. Thus, our results may provide evidence that the peripheral clock is disrupted at the entrance of active hypometabolism” (lines 372–376).

Reviewer #1

A striking and important finding of the current study is the death of B6N during 24h fasting under the lowest Ta and the absence of torpor under high Ta. These data may provide a significant clue what is needed to enter into or to ‘survive’ torpor. What do Tb and VO₂ patterns in these animals look like? Were they lower in body weight? When did they die? Is death due to not being able to maintain Tb, which may relate to the absence of shifting of Tr (a Q10 lowering of VO₂)? Or is death acutely at normal or slightly lower Tb? Similar questions arise for animals not entering torpor at higher temperatures.

Response

We are pleased to know that Reviewer #1 spotted one of the most interesting discoveries of this project. The torpor phenotype difference in very close inbred strains, B6N and B6J, is fascinating, and we were thrilled to report this in the current paper. We believe that it may be a lead to address the mechanisms of torpor induction. As a start, we have investigated the expression difference in various phases of torpor in B6J. The body weight and age do not differ in

these two groups (please see the table below). The only difference is the strain. When the B6N mouse dies without entering torpor, it is only when TA is lower than 20; we think B6N has less tolerance to low metabolism and low body temperature, and therefore some unknown mechanism is preventing the mouse from entering torpor and they die. When the TA is higher than 20, B6N mice do not enter torpor but they do not die at least in 24-hour fasting. The animals need to spend less energy when they stay in a warmer environment, as far as the TA is below the thermoneutral zone. We think B6N did not enter torpor at $TA > 20$ because they could not, and we guess they may die if we keep them fasted longer.

We added the following description in the discussion to emphasize our finding: “1) B6N enter torpor less than B6J, 2) the heat production sensitivity during torpor is higher in B6N than in B6J (Fig. 1). The age and body weight similarity used in this study (B6J, 8.00 ± 0.33 weeks-old, 22.75 ± 1.19 g, $n = 50$; B6N, 8.34 ± 0.53 weeks-old, 22.46 ± 1.25 g, $n = 46$) further emphasizes the distinct difference in torpor phenotypes. One possible explanation for the torpor phenotype difference is the potential difference in hypometabolism and hypothermia tolerance in these strains. In hibernators, their tissue shows cold tolerance even when they are not in hibernation^{49–51} suggesting the species-dependent level of cold tolerance. Therefore, it is possible that B6N and B6J have distinct hypometabolism or hypothermia tolerance.”(lines 347–354) To clarify the underlying mechanisms, we have decided to investigate these strains in detail in the future; however, we believe it is out of scope for this study.

Strain	Sex	BW (g)	Age (weeks)	n
C57BL/6J	female	17.71 ± 0.57	8.43 ± 0.15	8
C57BL/6J	male	22.75 ± 1.19	8.00 ± 0.33	50
C57BL/6NJcl	female	19.64 ± 0.75	8.67 ± 0.39	12
C57BL/6NJcl	male	22.46 ± 1.25	8.34 ± 0.53	46

Reviewer #1

Variation in torpor propensity does not necessarily depend on genetic factors, a common theme articulated in the manuscript, but may also involve epigenetic regulation (see e.g. doi: 10.1242/jeb.171983). Epigenetic changes, for instance DNA methylation of CpG in promoters, will also affect CAGE-Seq findings. Please point this out in the discussion.

Response

Thank you for a meaningful comment. We added the study presented by Reviewer #1 (doi: 10.1242/jeb.171983) in the reference and updated the discussion with a text mentioning that epigenetic changes may also affect the torpor phenotype and must be taken into consideration in future studies: “One thing to note is that torpor phenotype can be

affected by epigenetic changes such as the nutritional experiences during the fetal period⁸². Therefore, in future studies, epigenetic influences must be taken into consideration along with the SNP analysis” (lines 460-463). We agree that additional epigenetic data like Chip-seq or DNA methylation screening can improve our findings, but the final output will be the definition of genomic regulatory elements and transcription factors binding sites. Among numerous genome-wide and high throughput methods, CAGE is one of the most precise and sensitive. Also, using CAGE and genomic sequence, we can estimate the proportion of CpG islands involved in torpor because broad CAGE signal is a feature of CpG type promoters (Haberle, V., and Lenhard, B., 2016, *Semin. Cell Dev. Biol.*).

Reviewer #1

The authors define torpor as a drop in metabolic rate below 30% - this is a value not reached in the mice examined, notably B6J (e.g. line 117, Fig 2b). Consequently, many in the field will not regard the hypometabolism phase as torpor – this should be dealt with.

Response

We apologize that our original manuscript may be misinterpreted as torpor is defined as a 30% drop of VO₂. Rather, we defined torpor as when the animal exhibit lower values of both TB and VO₂ than the expected values from the normal condition. Every recording in this study uses the first 24-hour recording for estimation of normal condition with a 99.9 % of credible interval. We added several texts to the **Results** and the **Methods** section to make it clearer that our study uses torpor definition tuned for the individual animals (lines 89-90 and 540-550).

Reviewer #1

Minor
24: ‘is’ is lacking

We fixed it.

Reviewer #1

26: ATF3 is not highly expressed in liver, but in BAT

We fixed it.

Reviewer #1

37: superscript of 4; ‘As a result,....’ suppression of thermoregulation is not a result of hypometabolism (we don’t know)

To avoid confusion, we changed the sentence to “As a result, the homeostatic regulation of body temperature is

modified, and the total energy usage drops ” (line 38–39)

Reviewer #1

53: references 17 and 18 document epigenetic changes during hibernation (which might be outside promoters or CpGs), but do not prove strict epigenetic control

We changed to “Furthermore, several studies demonstrated epigenetic changes during hibernation^{17,18},” (line 51–52)

Reviewer #1

54: Similarly, reference 19 examines a poikilotherm marine animal quite distant from mammalian hibernators and both reference 19 and 20 do not prove ‘a role of MiRNAs’, but merely associated changes.

We changed a part of the sentence to “a relationship of miRNAs in the process” (line 52) and removed reference 19.

Reviewer #1

75: Many readers may not be familiar with ‘promotor shapes’ (at this point) -> distribution of transcription start sites within a promoter (or the like)

We rewrote ‘promotor shapes’ to “distribution of transcription start sites within a promoter.” (line 73–74)

Reviewer #1

90: may I suggest replace ‘lower than’ by: below the 99.9% CI (or the like)

We rewrote it as suggested.

Reviewer #1

147: form -> from

Fixed.

Reviewer #1

152: they show -> it shows

Fixed.

Reviewer #1

167: I think you cannot infer from the results that metabolism oscillation during torpor is an transcription-independent phenomenon; your sampling timing seems not apt and sample number too low.

Thank you for your fair comment. What we want to emphasize is the independence of transcriptome from the metabolic diversity during the torpor-capable period. Therefore the word ‘oscillation’ was removed, and the sentence was rewritten as: “indicating that the dynamic metabolic change during torpid episodes is based on a transcription-independent mechanism” (lines 173-174).

Reviewer #1

169: I do not get that you exclude circadian influences by experiment #2

This is our mistake. We removed the phrase ‘and exclude possible circadian rhythm effect.’

Reviewer #1

186: It seems that some of the most extremely changed dots in Fig. 2e are not annotated and represented in Fig. 2h,i. Can you explain?

Yes. The non-annotated dots are dots from unknown promoters. They have no name nor ID, therefore not written. Please look at Table_S1. ‘L2__chr11+_84173295’ is the right-bottom dot in Fig 2e.

Reviewer #1

202: the two essential -> one of the two essential

The sentence was fixed according to the suggestion.

Reviewer #1

223: please provide the number of promoters

The number was added.

Reviewer #1

229: both hunger and cold -> either hunger or cold

The sentence was fixed according to the suggestion.

Reviewer #1

259: In Fig. 4d, torpor-down is very similar to hypo-down. Yet in Fig. 4e only torpor down is shown. Please phrase an overall conclusion of Fig. 4d.

We think the promoter-shape similarity of torpor-down and hypo-down is a coincidence. Because our primary interest was to know the promoter shape in torpor-specific groups, we compared the torpor-down and torpor-up to the control

(All-promoters).

Reviewer #1

347: “reversible experiment”: not sure what you mean.

Response

We rewrote it to “*Arntl1* was up-regulated during torpor but not included in the hypometabolism-associated promoters”. (lines 368–369)

Reviewer #1

359: Sure there may be no change in AMPK subunits, yet pathway analyses consistently show it among the (top) regulated pathways; how to interpret this finding? AMP/ATP ratio – are regulated genes all downstream?

Response

Thank you for pointing out a meaningful issue. The discrepancy of AMPK expression and AMPK pathway enrichment is mainly due to PGC-1 α , FOXO1, and mTOR expression, as is shown in the figure below. The red genes are the differentially expressed genes. These are shreds of evidence for the unique nature of daily torpor muscle accompanied by the features from all three states; hibernation, starvation, and growth. Therefore, we added the following sentence: “Although, the enriched AMPK signaling pathway without AMPK expression bespeaks a complex nature of muscle transcription in daily torpor.” (lines 384–385)

Reviewer #1

Fig. S1d: the text B6N is behind the grid.

The sentence was fixed according to the suggestion.

Reviewer #2**General comments**

The studies by Deviatiiarov and colleagues aim to establish a difference in torpor between two closely related but genetically distinct mouse strains, and then exploit those differences to identify transcriptional start sites that might be involved in the regulation of torpor. Through various comparisons, the manuscript identifies 287 “torpor-specific” promoters in soleus muscle. The authors then focus on one gene, *Atf3*, which is involved in the stress response. Knocking out *Atf3* modestly weakens the torpor phenotype in mice.

It appears that a lot of data were generated in these studies, which will be of use to the scientific community. However, it is difficult to determine which experiments were performed at the same time. For the comparisons to be valid, it is essential that the treatment groups for each experiment have been analyzed concurrently and using comparable mice (of the same body weight, age, and sex). These variables are known to affect torpor but are not accounted for in this manuscript and may confound the major findings. These concerns would need to be addressed in order to substantiate the conclusions of the paper.

Reviewer #2

Major points:

1. The analyses of the torpor phenotype in BL/6J vs. BL/6N mice do not appear to be performed at the same time and appear to have different body weights. This means that the strain difference may be confounded by experimental conditions or body weight. This is a major finding of the paper, but the experimental design does not adequately support the conclusion.

Response

Thank you for pointing out a major issue in our manuscript.

The age and sex of animals used to compare B6J and B6N are matched as possible. The mean and standard deviations are summarized in the table, which was presented above in response to Reviewer #1.

Males were used mainly for comparison (Fig. 1b to 1j, and Sup. Fig. 1a to 1f). To confirm that torpor phenotypes differences are independent of sex, some females were tested in $T_a = 20\text{ }^{\circ}\text{C}$ (Sup. Fig. 1g and 1h). To note, 38 mice

in the B6J male data were reused from the previous study (Sunagawa, *Sci Rep*, 2016), as it is mentioned in the manuscript. We added the torpor recording data at $T_A = 28$ and 32 °C ($n = 12$), which is lacking from the previous dataset, and reanalyzed the body temperature characteristics during torpor in B6J male mice and compared it to B6N.

Reviewer #2

2. Lines 169-174 describe an analysis that was used to exclude possible circadian effects. The results are reported to be similar with respect to the VO₂ pattern and “a distinct transcriptome profile” but the promoters were not reported. Please provide a full analysis and discuss any differences in the results. Does the overlap among the different analyses change? Do other candidates arise?

Response

First, the circadian rhythm has nothing to do with this experimental setup. It is our mistake. Therefore, we removed the phrase ‘and exclude possible circadian rhythm effect’ from line 169.

Second, Experiment #2 was performed to test the reproducibility of CAGE analysis. The experiment for pre, mid, and post torpor was repeated twice. Data for the second run was shown in Sup. Fig. 2a and 2b. Transcriptional regulation is reproducible - we see clear clustering of pre- and post-torpor samples together, while all middle-torpor replicates are located in the separated cluster, highlighting the reversible activity of torpor-specific genes. To further clarify the robustness of this method, we analyzed the DE genes in Experiment #1 and #2. What we found was that 77-80 % of DE genes from Experiment #2 were included in the list of DE genes from Experiment #1, which was added in the results in the manuscript as: “To note, the reproducibility of this analysis was tested by comparing the DE genes found in experiments #1 and #2, which showed nearly 80% of genes overlapped.” (lines 195-196)

Reviewer #2

3. The analyses comparing HiT and ad lib fed mice to food-restricted “Mid” mice do not appear to be performed at the same time, which introduces many confounds. Additionally, the manuscript does not specify how do ad lib fed mice differ from “Pre” mice. Are the mice analyzed at similar body weight, sex, age, time of day?

Response

Thank you for pointing out the unclearness of our original manuscript. For the comparison of HiT, Fed and Mid mice, we admit that our methods are not perfect; however, we did our best to normalize the sampling. Not only of these three groups but all of the CAGE samples were collected from C57BL/6J male mice at the same age and at the same time of a day (ZT 22). The difference between “Pre” mice and “Fed” mice is the sampling day after their recording had started. Specifically, Pre mice were sampled at day 1, and Fed mice were sampled at day 2 (please see Fig. 2a)

of the experiment. The list below shows the age and bodyweight of the animals used for CAGE sampling. The animals were sampled on a different day because of the equipment limitation. Four animals at max could be recorded at once. Our experimental setup allowed a stable and reproducible result which can be inferred from the unbiased clustering results from the CAGE analysis. Please see Fig 2d and Fig Sup 2c. The Mid-group are taken in different experiments, but they form a single cluster. These results indicate that even the experiments are not performed at the same moment, the experimental conditions are sufficient to produce a comparable result.

Group	BW (g)	Age (weeks)	n
Pre	23.00 ± 0.88	8.57 ± 0.00	4
Mid	23.30 ± 1.09	8.14 ± 0.00	8
Post	23.45 ± 0.53	8.14 ± 0.00	4
Pre_2	24.20 ± 1.41	8.86 ± 0.00	2
Mid_2	24.02 ± 0.88	8.37 ± 0.33	5
Post_2	24.13 ± 1.37	8.57 ± 0.00	3
HiT	21.30 ± 0.51	8.29 ± 0.00	4
Fed	23.62 ± 0.45	8.71 ± 0.00	4
Deprived	22.25 ± 0.48	8.14 ± 0.00	4

Reviewer #2

4. Because FIT requires cold temperature and fasting, the idea to remove one variable is potentially informative. However, the design does not test for the isolated effect of fasting (which requires fed or fasted mice BOTH concurrently housed at high ambient temperature) or the effect of temperature (which requires two concurrent groups of fed mice housed at low or high ambient temperatures). As designed, the analysis is confounded by time, does not alter the dependent variables one at a time, and does not report information on other variables that are known to modulate torpor (e.g. body weight, sex, age). Additionally, this approach assumes that the effects of these variables is additive. Is there any evidence for that? At the very least, this should be explained as a major caveat.

Response

Thank you for a meaningful comment. The suggestions are all understandable. We would like to respond in the following way:

First, we agree that this experimental design cannot obtain the independent effect of fasting nor high temperature. Although, the aim of this experiment was to obtain the genes which show differential expression due to the torpid condition, not because the animal is fasting nor the animal is staying at a high-ambient temperature. By comparing HiT (Hi-temp + fasting) to the Mid (Lo-temp + fasting) group, we can tell what genes are expressed differently when

high-temperature prevents torpor induction. These results include the high-temperature DEs and the torpor-related DEs. In the same logic, by comparing Fed (Lo-temp + food) to Mid (Lo-temp + fasting), we can get the list of genes that are fasting related in addition to torpor related genes. Without the experiment with Hi-temp with food, it is impossible to tell what genes are related only with Hi-T or only with fasting, but it is out of our goal. Therefore, to sort out the genes related to torpor but not fasting nor hi-temp, the experiments are sufficient.

Second, the assumption that low-temperature and fasting are additive to torpor induction does not have evidence. What we can say from our results is that at least a 24-hour fasting and low TA (20 °C) can induce FIT, and without either of them, FIT will be prevented. We assume that even in a high TA if the animal fasted for a longer period, it may enter torpor. Or, even in a shorter period of fasting time, the animal may enter torpor when TA is lower. Therefore, we added the following description as a caveat of this approach: “One aspect to note is that either 24-hour fasting or low T_A is an essential factor in this experiment setup, but in other environments, such as in lower T_A or in longer fasting, removing the factors may not prevent torpor. Therefore, the hypometabolic factors will include the most genes related to torpor, but it still includes a fraction of genes that are related only to hunger or low T_A .”(lines 216–220).

Reviewer #2

5. The involvement of the Atf3 transcription factor in the regulation of torpor is unclear. The authors show that Atf3 expression is altered during torpor and that Atf3 may regulate torpor-specific promoters. However, it is unclear how the strain differences are explained by Atf3. Are there strain differences in Atf3 regulation or expression? What is the phenotype of the Atf3 KO mice beyond FIT? Are there body weight or other phenotypes that might confound the FIT response? The authors should also discuss how the function of Atf3 in the stress response might relate to torpor. How does stress alter FIT or torpor? Might the KO FIT phenotype reflect an interaction with stress?

Response

Thank you for the suggestion, which makes our manuscript clearer.

First, Atf3-KO tends to show a weaker phenotype of torpor, although there is no statistical significance in their difference. Because we have not analyzed the atf3 expression in B6N mice, it is impossible to infer that atf3 is responsible for the phenotype difference in B6J and B6N. Rather, this whole project has started to detect the genetic difference in these two strains, explaining the phenotype difference of torpor. What we found was a general difference in the transcriptional regulation in the skeletal muscles. We can only speculate that the strain difference stems from the difference in the torpor-specific genes, but it is not confirmed yet.

Second, how *Atf3* is involved in torpor regulation is not elucidated in this study. We have shown the correlation with torpor in skeletal muscle and demonstrated that systemic deletion of *Atf3* has a tendency of shallower torpor phenotype than wildtypes. *Atf3* is an adaptive-response gene, and that its expression is increased by numerous signals, including those triggered by genotoxic agents, cytokines, cell death-inducing agents, and physiological stresses (Ku, H.-C., 2020, *Front. Endocrinol.*). One widely accepted function of *Atf3* is that it plays an important role in dealing with ischemic/reperfusion injuries in various tissues (ref 70, 71, and 72 in the first submitted manuscript). Therefore, we hypothesized that *Atf3* is activated for ischemic tolerance for the tissue.

Third, the *Atf3*-KO did not show any obvious phenotype beyond FIT, as reported in the past literature (Hartman, M.G., 2004, *Mol. Cell. Biol.*). For example, the body weight did not show a clear difference according to the existence of the *Atf3* gene:

Exp	BW	Age	n
ATF3-021a wildtype	24.43 ± 2.77	9.28 ± 0.29	7
ATF3-021a hetero	24.56 ± 4.23	9.88 ± 0.81	7
ATF3-021a homo	25.67 ± 1.74	9.84 ± 0.89	7
ATF3-021b wildtype	27.10 ± 0.90	9.69 ± 0.56	6
ATF3-021b hetero	27.33 ± 2.34	10.57 ± 0.99	6
ATF3-021b homo	25.36 ± 1.67	9.51 ± 0.77	9
ATF3-025a wildtype	25.40 ± 1.82	10.80 ± 1.56	5
ATF3-025a hetero	26.59 ± 1.48	10.03 ± 1.14	5
ATF3-025a homo	26.00 ± 1.73	9.69 ± 1.22	5
ATF3-025b wildtype	25.81 ± 1.17	10.83 ± 1.34	6
ATF3-025b hetero	24.83 ± 1.39	10.46 ± 1.01	6
ATF3-025b homo	25.23 ± 1.80	10.69 ± 0.94	6

In respect of Reviewer #2's suggestion, we have added discussions of how the function of *Atf3* in the stress response might relate to torpor (lines 427–454). We raised two hypotheses: 1) *atf3* is expressed for preparation of ischemia/reperfusion injuries, 2) *atf3* is expressed for preparation for glucose depletion.

Reviewer #2

Minor points:

1. The manuscript does not adequately explain H or T(R) (lines 120-122). It is insufficient to cite the previous publication. Please explain the variables, what they indicate about the physiology of torpor, and, most importantly,

what is the implication of a strain difference in H?

Response

The explanation of H and TR is added at lines 119-124 as: “The thermogenesis system of a mammal can be modeled as a simple negative feedback system with a set-point temperature (or a reference temperature) T_R and the heat production sensitivity or the negative feedback gain H^4 . Under a steady-state condition with most VO_2 consumed for heat production, VO_2 equals to $(T_R - T_B) H$. This equation is not affected by T_A ; therefore, recording T_B and VO_2 at different T_A s can provide an estimation of H and T_R . In this way, H and T_R of B6N from the VO_2 and T_B were estimated.”. Additionally, what H difference implies is added to the results as: “Higher H implies stronger feedback against the T_R to T_B gap. Generally, H becomes lower during torpor. Higher H in B6N during torpor indicates B6N has the ability to reach closer to T_R during torpor, which can be explained as B6N is less adapted to hypothermia/hypometabolism than B6J”(lines 132-135)

Reviewer #2

2. The abstract and first sentence of the introduction mention that VO_2 drops to <30% of baseline. It does not appear that any of the FIT responses in the main figures reach this threshold. The definition of torpor should be consistent in the text and figures (and the literature).

Response

Thank you for pointing out an unclear statement we have made. Some mouse shows VO_2 drop less than 30%, especially when the ambient temperature is as low as 8 °C. Please look at Sup Fig 1b or 1h. But this is not true for every mouse. So we have updated the abstract and first sentence of the introduction to make it clearer that not every but in certain conditions the animal can show VO_2 less than 30%.

In our study, torpor is defined as when the animal exhibit lower values of both T_B and VO_2 than the expected values from the normal condition. Every recording in this study used the first 24-hour recording as the normal condition of the animal and estimated the lower limit of the normal condition for each animal. We added several texts to **Results** and to **Methods** section to make it clear that our study uses torpor definition tuned for the individual animals (lines 89-94 and 544-545).

Reviewer #2

3. In line 41, “resistance” appears twice. Perhaps “tolerance” would be more appropriate because the mice do not resist (instead they actively induce) the decreases in body temperature or oxygen consumption.

Response

We agree that ‘tolerance’ connotes the notion better than ‘resistance.’ We replaced both words with ‘tolerance.’

Reviewer #2

4. Line 47 mentions a2-macroglobulin and cites previous work. This molecule is never mentioned again. Was it identified in any of the current studies?

Response

No. We brought this up to introduce the history of differential gene expression study in hibernators. Although, as you felt, referring to this manuscript specifically may confuse the readers. Therefore, we removed the current sentence and merged the reference in the following sentence: “..., series of transcriptomic investigations were conducted in well-studied hibernating animals, including ground squirrels, ... (lines 47–48)”.

Reviewer #2

5. Line 64 claims that FIT was developed by the authors. Please see “Fasting-induced torpor in *Mus musculus* and its implications in the use of murine models for human obesity studies” published by Webb, Jagot, and Jakobson in 1982.

Response

We apologize that our original manuscript can mislead the reader as we have developed FIT. We understand that FIT was documented or found in past studies. What we have done in ref 4 (Sunagawa, *Sci. Rep.*, 2016) is that developed a stable method to initiate and detect torpor in mice. Even in the Webb, Jagot, and Jakobson paper in 1982, not all the mouse is recognized to enter torpor. In ref 4, the major finding was in B6J mice, exploiting the newly developed detection method, all of the animals enter torpor when removing food for 24 hours from ZT-0, in a wide range of ambient temperature. To make this clear, we updated line 64 as follows and added the reference to the Webb paper with respect: “Notably, the mouse is well-known to enter torpor by fasting^{28,29}, and we recently developed an improved method to initiate and detect fasting-induced torpor (FIT) in mice reproducibly⁴, ...” (lines 61-63)

Reviewers' comments:

Reviewer #1 (Remarks to the Author):

The authors have courteously addressed my comments and many of the unclarities have been adequately dealt with.

My main issue, however, remains. The CAGE-Seq data and Atf3 KO experiments in B6J are hard to connect to the B6N experiments. I acknowledge the effort in the authors having conducted SNP analysis, yet in itself (even when SNPs would have been identified) this is insufficient proof to attribute the difference in torpor phenotype between the strains to genetic factor(s). I feel the paper qualifies even without B6N data.

Secondly, it is still unclear to me to what extent Atf3 KO affects torpor behavior and a formal analysis is absent. I realize this is difficult given the differential effect on the 4 strains, yet seems mandatory to substantiate the claim that "The weakened torpor phenotype of Atf3-KO mice supports the torpor-related function of ATF3".

Finally, when do B6N mice receiving FIT at Ta 8-12 °C die (Fig 1c)? Do they engage in torpor and then succumb? Is this then towards the end of the bout? Or do they demise when (expected) to enter torpor, or otherwise?

Reviewer #2 (Remarks to the Author):

The authors have changed the manuscript to some degree in response to reviewer comments. I appreciate the clarifications made in the revision, fixing confusing statements or improving clarity. In the response to reviewers, the authors admit that multiple experiments are "not perfect" but do not do enough to mitigate experimental weaknesses nor do discuss how weaknesses affect the interpretations. I do not believe that the central claims of the paper are adequately substantiated by the analyses. The paper claims to compare torpor and transcriptional regulation in B6N vs. B6J, but it only tested torpor in B6N and transcriptional regulation in B6J.

Reviewer 1 suggested either CAGE-seq on B6N or "the authors may choose to leave out or reduce the sections dealing with B6N." The authors decline to add the B6N CAGE-seq data but do not reduce the sections dealing with B6N. I disagree with the authors that the comparison of B6N and B6J CAGE-seq data is beyond the scope of the paper because the paper is currently written as a comparison of transcriptional regulation between the two strains. If the B6N data will not be generated, then CAGE-seq data should be interpreted for B6J, not the comparison. Please note the strain in the Figure Legend.

I also agree with reviewer 1 that some readers will disagree that the hypometabolic phase qualifies as torpor. The revision explains the definition used with more detail but does not fully justify why it should be considered torpor.

The authors cite as a major issue that the hypometabolic phenotypes of B6J vs. B6N were not studied at the same time. This is a major limitation. The authors provide a table in the response to reviewers but neglect to provide all of the information within the manuscript. The revised manuscript only includes the mean \pm SE \pm SD? for males. I could not find the information for females, which do differ in their body weight, anywhere in the manuscript or supplementary files. Additionally, the fact that the studies were not concurrent is a major caveat of the study that must be dealt with in the discussion.

The statistical details should be more prominently displayed in the figures and text to avoid misleading

the reader. Lines 334-336 are misleading: "When the gene is deleted systemically, the animals tend to show a weaker phenotype of torpor, which supports that ATF3 is important for fasting-induced torpor regulation." Instead, it should be explicitly stated that the difference was not significant. The authors suggest, "how Atf3 is involved in torpor regulation is not elucidated in this study." However, the lack of a statically significant effect of Atf3KO on hypometabolism suggests that a role for Atf3 in hypometabolism should be excluded. There is not sufficient evidence to reject the null hypothesis that Atf3 is not involved in hypometabolism. It is not appropriate to suggest that it may have a role or speculate how it might have a role.

The authors report that "the Atf3-KO did not show any obvious phenotype beyond FIT, as reported in the past literature (Hartman, M.G., 2004, Mol. Cell. Biol.). For example, the body weight did not show a clear difference according to the existence of the Atf3 gene" Please provide your data showing that body weight or other phenotypes are not affected. These should be a part of the manuscript, not just the response to reviewers.

Finally, it is odd that the Hrvatin 2020 paper is not cited.

Reviewer Comments, Author Responses, and Manuscript Changes

(in response to COMMSBIO-20-3364A)

The box is the comments from the reviewers.

Green letters denote the updated text in the current revision.

Reviewer #1

My main issue, however, remains. The CAGE-Seq data and Atf3 KO experiments in B6J are hard to connect to the B6N experiments. I acknowledge the effort in the authors having conducted SNP analysis, yet in itself (even when SNPs would have been identified) this is insufficient proof to attribute the difference in torpor phenotype between the strains to genetic factor(s). I feel the paper qualifies even without B6N data.

Response

We sincerely thank you for acknowledging the efforts somehow to tie B6N data to the B6J CAGE-Seq data. Although, we admit that the approach we have taken in this paper is not adequate to bind the B6N FIT phenotype to that of B6J. Therefore, as you have suggested initially, we decided to remove the B6N data from this paper. The following are the major changes we have made related to the B6N experiments.

- Lines 78 - 149 were deleted.
- Fig. 1 and Supplementary Fig. 1 were removed except Sup Fig 1e, which was moved to the first figure in the new Sup Fig 2.
- Sup Fig 4f and 4g were removed.

Reviewer #1

Secondly, it is still unclear to me to what extent Atf3 KO affects torpor behavior and a formal analysis is absent. I realize this is difficult given the differential effect on the 4 strains, yet seems mandatory to substantiate the claim that “The weakened torpor phenotype of Atf3-KO mice supports the torpor-related function of ATF3” .

Response

Thank you for pointing out the weakness of our analysis. As you see, not every strain showed a weakened phenotype of torpor compared to wildtype B6Js. Because all four strains are siblings from hetero parents, we have decided to estimate the difference of torpor phenotype within the KO gene and compared how the hetero and homo allele KO strains differ from the wildtype mice. We made a Bayesian statistical model that includes the phenotype difference and estimated the parameters to evaluate the difference.

The results show ATF3-021a has higher minimal TB in hetero- or homo-KOs (new Supplementary Fig. 4f). The VO2

in the same animals did not show a significant difference. ATF3-025b showed lower TB than wildtypes. The other two strains did not show differences from wildtype animals. It is difficult to understand how ATF3 is regulating the FIT phenotype only from our KO studies, although, at least, we can say that deleting Atf3 may have a torpor-related function. According to the analysis, we added a discussion as “One thing to note is the dissimilar phenotype of two Atf3 knockout strains, ATF3-21a and ATF3-25b. These results indicate a clear yet complicated relationship of Atf3 to torpor regulation, and further studies are mandatory for clarification. (new lines 335-338)” .

Reviewer #1

Finally, when do B6N mice receiving FIT at Ta 8-12 oC die (Fig 1c)? Do they engage in torpor and then succumb? Is this then towards the end of the bout? Or do they demise when (expected) to enter torpor, or otherwise?

Response

Although we have not quantitatively analyzed the dying moment of B6N, many of the animals seem to die after they enter torpor. We are eager to address the underlying mechanisms of the B6N-specific torpor-incapability in the future.

Reviewer #2

The authors have changed the manuscript to some degree in response to reviewer comments. I appreciate the clarifications made in the revision, fixing confusing statements or improving clarity. In the response to reviewers, the authors admit that multiple experiments are “not perfect” but do not do enough to mitigate experimental weaknesses nor do discuss how weaknesses affect the interpretations. I do not believe that the central claims of the paper are adequately substantiated by the analyses. The paper claims to compare torpor and transcriptional regulation in B6N vs. B6J, but it only tested torpor in B6N and transcriptional regulation in B6J.

Reviewer 1 suggested either CAGE-seq on B6N or “the authors may choose to leave out or reduce the sections dealing with B6N.” The authors decline to add the B6N CAGE-seq data but do not reduce the sections dealing with B6N. I disagree with the authors that the comparison of B6N and B6J CAGE-seq data is beyond the scope of the paper because the paper is currently written as a comparison of transcriptional regulation between the two strains. If the B6N data will not be generated, then CAGE-seq data should be interpreted for B6J, not the comparison. Please note the strain in the Figure Legend.

Response

We thoroughly admit that the data we have obtained so far is insufficient to relate the B6N FIT phenotype to B6J. Therefore, we decided to remove the B6N data from this manuscript and hugely reform the manuscript's structure. The major deletion is listed above in the responses to reviewer #1.

Reviewer #2

I also agree with reviewer 1 that some readers will disagree that the hypometabolic phase qualifies as torpor. The revision explains the definition used with more detail but does not fully justify why it should be considered torpor.

Response

Thank you for mentioning yet another point that might be unclear to the readers. We have added an explanation of the torpor definition in the methods as “For each animal, TB and VO₂ were evaluated every six minutes for three days. From the recordings of day 1, the baseline metabolism was estimated with a certain credible interval (CI). In this study, we used the 99.9% CI of the posterior distribution of the estimated metabolism to detect outliers (new lines 419-422).”

Reviewer #2

The authors cite as a major issue that the hypometabolic phenotypes of B6J vs. B6N were not studied at the same time. This is a major limitation. The authors provide a table in the response to reviewers but neglect to provide all of the information within the manuscript. The revised manuscript only includes the mean \pm SE/SD for males. I could not find the information for females, which do differ in their body weight, anywhere in the manuscript or supplementary files. Additionally, the fact that the studies were not concurrent is a major caveat of the study that must be dealt with in the discussion.

Response

Because the B6N data was removed in the current revised manuscript, the comparison of B6N females to B6J females lost its value. Therefore, the female data was removed. Although, we thank Reviewer 2 for pointing out the importance of body weight in torpor experiments. We realized that female data were only written in the methods section apart from male data, which would be inconvenient to read. We will take the suggestion seriously for future studies and pay an effort to express body weight data much more apparent in the upcoming manuscripts.

Reviewer #2

The statistical details should be more prominently displayed in the figures and text to avoid misleading the reader. Lines 334-336 are misleading: “When the gene is deleted systemically, the animals tend to show a weaker phenotype of torpor, which supports that ATF3 is important for fasting-induced torpor regulation.” Instead, it should be explicitly stated that the difference was not significant. The authors suggest, “how Atf3 is involved in torpor regulation is not elucidated in this study.” However, the lack of a statically significant effect of Atf3KO on hypometabolism suggests that a role for Atf3 in hypometabolism should be excluded. There is not sufficient evidence to reject the null hypothesis that Atf3 is not involved in hypometabolism. It is not appropriate to suggest that it may

have a role or speculate how it might have a role.

Response

Thank you so much for pointing out the central weak point of our manuscript. We agree that it is not proper to state “the animals tend to show a weaker phenotype of torpor” without statistical analysis. We decided to analyze the minimal body temperature and VO₂ of the Atf3-KOs to clarify how the deletion of Atf3 may affect the FIT phenotype. Especially, to focus on the effect of the target gene, we hypothesized that KO animals have additive torpor phenotype difference from the wildtype torpor phenotype and generated a model that includes a parameter for the difference from the wildtype phenotype. Applying Bayesian statistics, we estimated the minimal TB and VO₂ difference from the wildtype animal for each KO, namely 021a, 021b, 025a, and 025b.

As we have mentioned above, 021a showed a significant lowering in minimal TB during torpor. Interestingly, 025b showed a deeper phenotype during torpor. Collectively, at least one out of four Atf3-KO strains has shown a weakened phenotype of torpor. One out of four Atf3-KO showed an opposite phenotype from the original KO screening. We do not have enough data to explain why an opposite phenotype is generated when the same gene is knocked out. What we can say is that two Atf3-KO, namely Atf3-021a and Atf3-025b, affect FIT phenotype. We added the new analysis results (new lines 233-240, 242-245) and updated the discussion (new lines 335-338).

Reviewer #2

The authors report that “the Atf3-KO did not show any obvious phenotype beyond FIT, as reported in the past literature (Hartman, M.G., 2004, Mol. Cell. Biol.). For example, the body weight did not show a clear difference according to the existence of the Atf3 gene” Please provide your data showing that body weight or other phenotypes are not affected. These should be a part of the manuscript, not just the response to reviewers.

Response

We truly apologize that we did not take your suggestion as you intended. We have added the information of the Atf3-KO bodyweights as a chart in Supplementary Fig. 4e. We did not run any other specific tests for other phenotypes. That is why we wrote, “the Atf3-KO did not show any obvious phenotype beyond FIT”.

Reviewer #2

Finally, it is odd that the Hrvatin 2020 paper is not cited.

Response

Thank you for pointing out a critical study that we should not have missed. We added the citation to the introduction as “Furthermore, one group identified neurons which are regulating the induction of FIT, and our group identified genetically labeled neurons which can induce a hibernation-like state in mice. (new lines 58-60)” .

Reviewers' comments:

Reviewer #1 (Remarks to the Author):

The manuscript by Deviatiiarov et al. has been considerably adapted to outline a coherent story on their usage of CAGE-seq to identify skeletal muscle genes by that are specifically affected by the hypometabolic component of mouse torpor, taking advantage of the 'rich genetic resources' in mice. In torpid mouse muscle they found a large set of DE genes between pre-torpor, torpid and post-torpor mice sampled at identical timepoints, largely corresponding to those previously found in seasonal hibernators. Next, the authors sought to and compared DE in genes from torpid animals with low ambient temperature adlib fed mice, fasted animals on high ambient temperature and mice in which torpor was precluded by a tactile stimulus. They focussed on genes that were DE in torpor and even to a greater extent in 'precluded torpor' from which they distilled Atf3 as a principal overshoot DE gene. Following CRISPR/Cas9 knockout they explored the effect of constitutive Atf3 KO on torpor propensity and essentially found no difference with wt.

Major remarks

1. The Atf3 KO experiments and conclusions hinge on the assumption that tactile stimulation precluding torpor actually increases torpor propensity. This to my knowledge is unsupported by literature and the current paper should provide the data. Absence of an effect of tactile stimulation on torpor propensity may well explain the lack of robust effect, if any, of Atf3 KO on torpor. Notably, Atf3 is upregulated by a variety of stress signals, which may also explain its overshoot in torpor-precluded mice.

Further, the data set shows 2 Atf3 promoters detected. Whereas both are upregulated in torpor precluded mice, only one is DE is a torpor-specific, hypometabolic gene. Please explain.

2. At several occasions, claims are made regarding difference in promotor usage ("promoter shift" (l. 68), "alternative promoter usage" (l. 125)), yet the sections do not provide underpinning data. The authors also report a difference in shape index between downregulated torpor-specific promoters (but not upregulated!) and all promoters. Also, the discussion on its potential origin (l. 307-309) is rather limited and unsupported by literature. The authors may consider an advanced analysis to substantiate their claim on the advantage of using mouse in view of its genetic resources.

Minor

l. 25 and others: there is confusion on the total number of promoters: both 12,682 and 12,683 are reported.

l. 62,63: are remnants of a previous version; please redefine.

l. 88,89: the statement is confusing. As it reads, 1,729 (13%) TSS were not mapped to genes. Was this really the case, and if so how to view this in the light of the 'rich genetic resources'?

l. 97,98: your data cannot substantiate or refute an independence of metabolic switches in torpor from transcription.

l. 119: what is meant by the statement that '80% overlapped'? Were non-overlapping genes present in both sets? Were number of promoters detected similar? What DE gene comparison is referred to, the sum-fold change?

Reviewer Comments, Author Responses, and Manuscript Changes

(in response to COMMSBIO-20-3364B)

The box is the comments from the reviewers.

Red letters denote the updated text in the current revision.

Dear Reviewer #1,

We deeply thank you for your enthusiasm returning us plenty of constructive comments every time. Your comments have improved our manuscript dramatically, and we hope the current version appeases you.

Reviewer #1

1. The Atf3 KO experiments and conclusions hinge on the assumption that tactile stimulation precluding torpor actually increases torpor propensity. This to my knowledge is unsupported by literature and the current paper should provide the data. Absence of an effect of tactile stimulation on torpor propensity may well explain the lack of robust effect, if any, of Atf3 KO on torpor. Notably, Atf3 is upregulated by a variety of stress signals, which may also explain its overshoot in torpor-precluded mice.

Response

Thank you for pointing out the limitation of our study. We agree that there are no previous papers stating tactile stimulation increases torpor propensity. We did not run an independent experiment for recording metabolism data of tactically stimulated mice. However, when performing the stimulation on-site, we often observed mouse trying to drop their metabolism (VO₂) when we stopped stimulating. It very much looked similar to sleep-deprived mice except that they had fasted. Therefore, we explained this fact in the newly added limitation section in the discussion, as well as the possibility of the Atf3 upregulation due to a variety of stress signals as: “**One limitation of this study is that Atf3 was identified through the torpor precluding study by tactile stimulation. A sleep deprivation study inspired this method. To our knowledge, this is the first time to report torpor deprivation in mice. Even we have frequently observed animals trying to lower their metabolism unless we touch them during the experiment, we do not have a quantitative test whether the torpor propensity will increase by this procedure. Therefore, the higher expression of Atf3 in torpor-deprived animals could be explained simply by the nature of this gene as a stress-inducible gene. Moreover, little effect to torpor-debt may explain the lack of robust effect to the torpor of Atf3 knockout animals.** (new lines 355–362)”.

Reviewer #1

Further, the data set shows 2 Atf3 promoters detected. Whereas both are upregulated in torpor precluded mice, only one is DE is a torpor-specific, hypometabolic gene. Please explain.

Response

Thank you for a perceptive observation. Indeed, both of the promoters detected in our data are documented as P1 and P2 in mammal Atf3 (Miyazaki, K., et al, Nucleic Acids Res. 37, 1438–1451, 2009). In mice, P1 stays 34.7 kb upstream from the canonical promoter site P2. The one we focused on is P2, which was torpor-specific and upregulated in torpor precluded mice. P1, which was upregulated in torpor-deprived mice but not being a torpor-specific gene, is reported to be the promoter site for a constitutively active Atf3 expression in some human cancer cell lines. This indicates even the same proteins are produced from the transcripts, and the regulation should be different. In this study, we have deleted the protein-coding region of Atf3 to test the function to torpor. There is, however, a possibility that deletion of the protein itself may damage the endogenous function of the P1 promoter-driven transcript, which has nothing to do with torpor phenotypes. Therefore, we have added descriptions about the two Atf3 promoters in the manuscript. Specifically, in the results section, we added: “Atf3 has two documented promoters³⁶. This study detected the canonical promoter as a torpor-specific promoter and an up-regulated promoter in a torpor deprived animal. Transcripts from the other promoter located 34.7 kbp upstream from the canonical promoter did not change significantly in this study. (new lines 217–221)”, and in the results section, we added: “Furthermore, the Atf3 promoter detected in this study is one of the two documented promoters of Atf3³⁶. Even the same proteins are produced from the transcripts, deleting the shared protein-coding region may affect the endogenous function of the Atf3 transcript from the other promoter. To clarify this, tweaking the promoter region to evaluate the two distinct transcripts would be necessary independently. (new lines 348–352)”.

Reviewer #1

2. At several occasions, claims are made regarding difference in promotor usage (“promoter shift” (l. 68), “alternative promoter usage” (l. 125)), yet the sections do not provide underpinning data. The authors also report a difference in shape index between downregulated torpor-specific promoters (but not upregulated!) and all promoters. Also, the discussion on its potential origin (l. 307-309) is rather limited and unsupported by literature. The authors may consider an advanced analysis to substantiate their claim on the advantage of using mouse in view of its genetic resources.

Response

Thank you for bringing up an unclear description in our manuscript.

For the term “promoter shift”, we have changed it to “shape index”, because in this study analysis of individual TSS shifts are not available currently. An additional analysis of CAGE TSS peaks within promoter regions is required.

A number of alternative promoters are available from Table_S1—316 genes that have alternative promoters in our data—having two or more transcripts associated with CAGE peaks. In our study, we discovered CAGE peaks far

outside from known promoter regions (inter or intragenic), which could represent real promoters of low expressed genes, enhancer RNAs, or side effects of CAGE protocol. To avoid false-positive cases we focused on CAGE peaks located within known promoter regions. In this study, each CAGE peak within promoter regions was associated with one specific transcript highlighting the richness of existing annotations in mice.

The shape index results are connected to the promoter structure and uncover regulation on alternative TSSs. Here we see increased index in down-regulated promoters, which leads to the sharp CAGE peaks in torpor conditions and could be related to inactivation of alternative TSSs and one TSS remaining under torpor. Stable index in up-regulated genes means that alternative CAGE TSS don't show critical changes in its expression. The index defines general trends of alternative TSS usage in torpor adaptation, while the list of specific genes and Zenbu viewer provide sufficient information for manual curation of individual genes.

Reviewer #1

Minor

1. 25 and others: there is confusion on the total number of promoters: both 12,682 and 12,683 are reported.

Response

We apologize for the confusion. The correct total number of the promoters is 12,862; therefore, we have updated two numbers from 12,863 to 12,862 (new lines 112 and 157).

Moreover, during this revision, we found the following two errata in our manuscript, which both of them were fixed.

1. The promoter at “L2__chr4_-_86669995” in Table_S1.xlsx is duplicated. One row was deleted.
2. The total number of unique genes is 10,617 instead of 10,615 (new line 89).

Reviewer #1

1. 62,63: are remnants of a previous version; please redefine.

Response

The text was updated as “**This study aimed to analyze the comprehensive gene expression at the skeletal muscle by introducing mice as a model for active hypometabolism,** (new lines 63–64)”.

Reviewer #1

1. 88,89: the statement is confusing. As it reads, 1,729 (13%) TSS were not mapped to genes. Was this really the case, and if so how to view this in the light of the ‘rich genetic resources’

Response

These 1,729 CAGE peaks were unable to connect with existing gene models by the +/-500 bp rule. It means these

peaks are located in intergenic or intragenic regions and may represent a transcriptional activity or related to unwanted side effects of the method. In our work, we focus on CAGE peaks located in known promoter regions. These new CAGE peaks are also available from Table_S1 for future manual curation. We added the following text in the results section: “, and the remaining was out of ± 500 -bp regions from the 5' ends of annotated genes (new line 89)”.

Reviewer #1

l. 97,98: your data cannot substantiate or refute an independence of metabolic switches in torpor from transcription.

Response

Thank you for pointing out an overstatement. We phrased it much objective as: “**indicating that the oscillating metabolic change during torpor does not show a clear difference in transcription** (new lines 97–98)”.

Reviewer #1

l. 119: what is meant by the statement that ‘80% overlapped’? Were non-overlapping genes present in both sets? Were number of promoters detected similar? What DE gene comparison is referred to, the sum-fold change?

Response

To show reproducibility of the results, we performed CAGE analysis on two independent sets of mice which were sequenced in separate batches. The number of promoters that could be detected was different in the sets because of the sequencing depth and the number of samples (9,311 and 9,916). However, we showed that despite these conditions for both of the mice groups, similar differentially expressed (DE) genes could be found in Pre vs. Mid and Mid vs. Post comparisons, which overlapped 80.17 % and 76.82 %, respectively. We added the text to clarify the comparison (new lines 117–119). DE genes were defined by standard edgeR protocol (FDR < 0.05).